# GIR-BENCH: VERSATILE BENCHMARK FOR GENERATING IMAGES WITH REASONING

**Hongxiang Li[1,4*], Yaowei Li[2*], Bin Lin[2], Yuwei Niu[2], Yuhang Yang[3],**
**Xiaoshuang Huang[4], Jiayin Cai[4], Xiaolong Jiang[4], Yao Hu[4], Long Chen[1†]**

[1] The Hong Kong University of Science and Technology    [2] Peking University
[3] University of Science and Technology of China    [4] Xiaohongshu Inc.

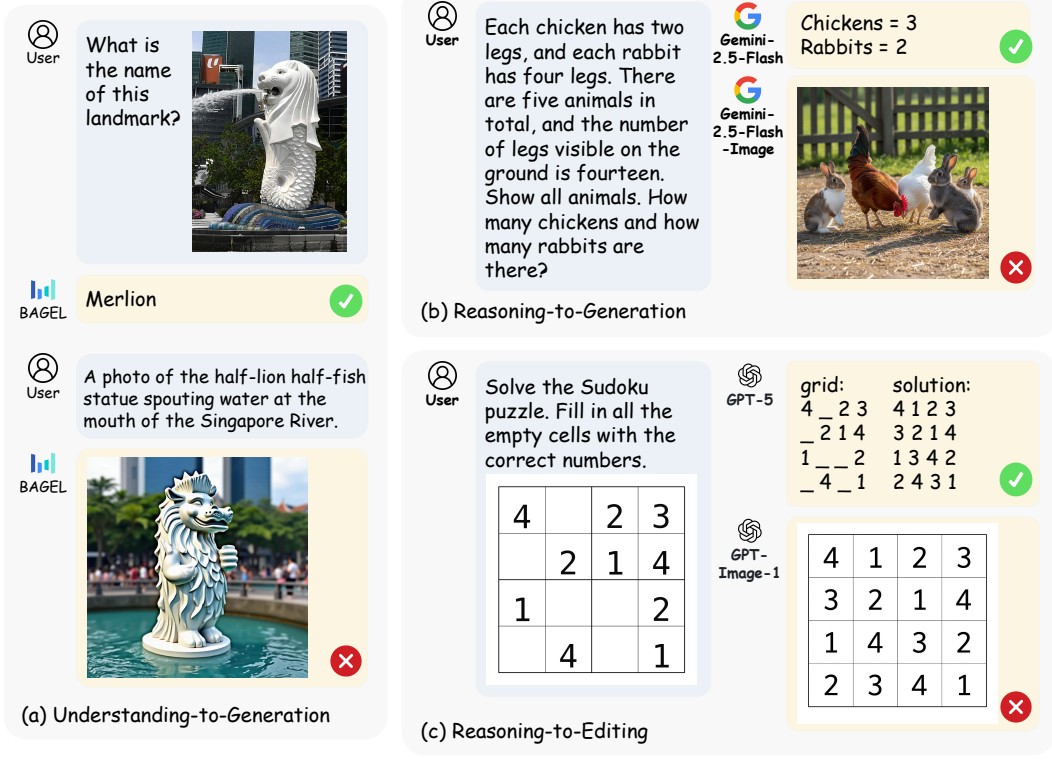

Figure 1: Illustration examples of GIR-Bench, which highlight misalignments between the reasoning and generation capabilities of state-of-the-art unified multimodal models.

## ABSTRACT

Unified multimodal models integrate the reasoning capacity of large language models with both image understanding and generation, showing great promise for advanced multimodal intelligence. However, the community still lacks a rigorous reasoning-centric benchmark to systematically evaluate the alignment between understanding and generation, and their generalization potential in complex visual tasks. To this end, we introduce **GIR-Bench**, a comprehensive benchmark that evaluates unified models across three complementary perspectives. Firstly, we investigate understanding–generation consistency (GIR-Bench-UGC), asking whether models can consistently leverage the same knowledge in both understanding and generation tasks. Secondly, we investigate whether models can perform reasoning-centric text-to-image generation that requires applying logi-

---

* Equal contribution.
† Corresponding author.

cal constraints and implicit knowledge to generate faithful visual content (GIR-Bench-T2I). Thirdly, we evaluate whether models can handle multi-step reasoning in editing (GIR-Bench-Edit). For each subset, we carefully design different task-specific evaluation pipelines tailored for each task. This enables fine-grained and interpretable evaluation while mitigating biases from the prevalent MLLM-as-a-Judge paradigm. Extensive ablations over various unified models and generation-only systems have shown that: Although unified models are more capable of reasoning-driven visual tasks, they still exhibit a persistent gap between understanding and generation. The data and code for GIR-Bench are available at https://github.com/HKUST-LongGroup/GIR-Bench.

# 1 INTRODUCTION

Image generation and editing techniques (Stability AI, 2024; Labs, 2024; Batifol et al., 2025; Liu et al., 2025) have evolved rapidly, demonstrating strong capabilities in producing high-quality visual content aligned with explicit prompts. However, existing models still struggle with complex visual generation tasks that require multi-step reasoning. This limitation has triggered the research focus toward unified multimodal models, where a single model supports both image generation and understanding. By leveraging the intrinsic knowledge and reasoning abilities of multimodal large language models (MLLM), such unified approaches enable richer expressiveness and more controllable image generation. Recent breakthroughs, such as Gemini-2.5-Flash-Image (DeepMind, 2025) and GPT-Image (OpenAI, 2025b), further highlight the transformative potential of this paradigm, showing that unified models can fundamentally reshape real-world applications, empowering users to accomplish complex visual tasks through natural language interaction. Compared to generation-only models, unified models (Deng et al., 2025; Chen et al., 2025a; Xie et al., 2025b) promise substantial gains. With these advances comes a critical **research question**: *how do we comprehensively evaluate the extent to which enhanced multimodal understanding improves generation capabilities?*

Earlier image generation benchmarks (Ghosh et al., 2023; Huang et al., 2023; Hu et al., 2024) mainly focused on object attributes and compositional evaluation, but remained limited to shallow mappings between text and visual content. Recent works (Zhao et al., 2025; Niu et al., 2025; Sun et al., 2025) have attempted to consider the reasoning capability. However, existing benchmarks remain limited in both their evaluation dimensions and protocols, making them inadequate for capturing the full capabilities of unified models.

- **For evaluation dimensions**, they cannot quantify the alignment between reasoning and generation within unified models. As shown in Figure 1, we observe consistent misalignments in transferring knowledge to generation, reasoning to generation, and reasoning to editing. For instance, a unified model may correctly recognize a real-world entity (*e.g.,* the Merlion in Figure 1), but still fail to generate it with an explicit description. Revealing and quantifying the gap is crucial, as it not only uncovers the inherent limitations of current unified models but also verifies whether they can deploy their knowledge and reasoning abilities equally across understanding and generation tasks.
- **For evaluation protocols**, prevalent benchmarks always propose some challenging tasks such as idiom interpretation to investigate unified models, but they cannot decompose these designed reasoning-centric tasks into more evaluable and interpretable forms. Instead, they rely heavily on the MLLM-as-a-Judge paradigm, where MLLMs serve directly as evaluators and evaluation scores are obtained through visual question answering. Such dependence inevitably couples the evaluation result with the biases and limitations of multimodal models themselves.

To this end, we introduce **GIR-Bench**, a benchmark designed to systematically explore the capability boundaries of unified models in reasoning-driven image generation and editing. The benchmark consists of three components: 1) **GIR-Bench-UGC**: To assess understanding–generation consistency (UGC), we evaluate whether models can reliably leverage the same knowledge for both recognizing and generating real-world entities. Specifically, we compile 300 entities from zoology, botany, and geography, design implicit reasoning-oriented prompts, and pair them with curated reference images (Figure 1(a)). This design allows us to systematically measure the consistency gap between understanding and generation. 2) **GIR-Bench-T2I**: We then explore reasoning-centric text-to-image generation (*c.f.,* Figure 1(b)). It requires models not only to retrieve relevant knowledge but also to apply precise logical reasoning to faithfully satisfy specified constraints. We design 300

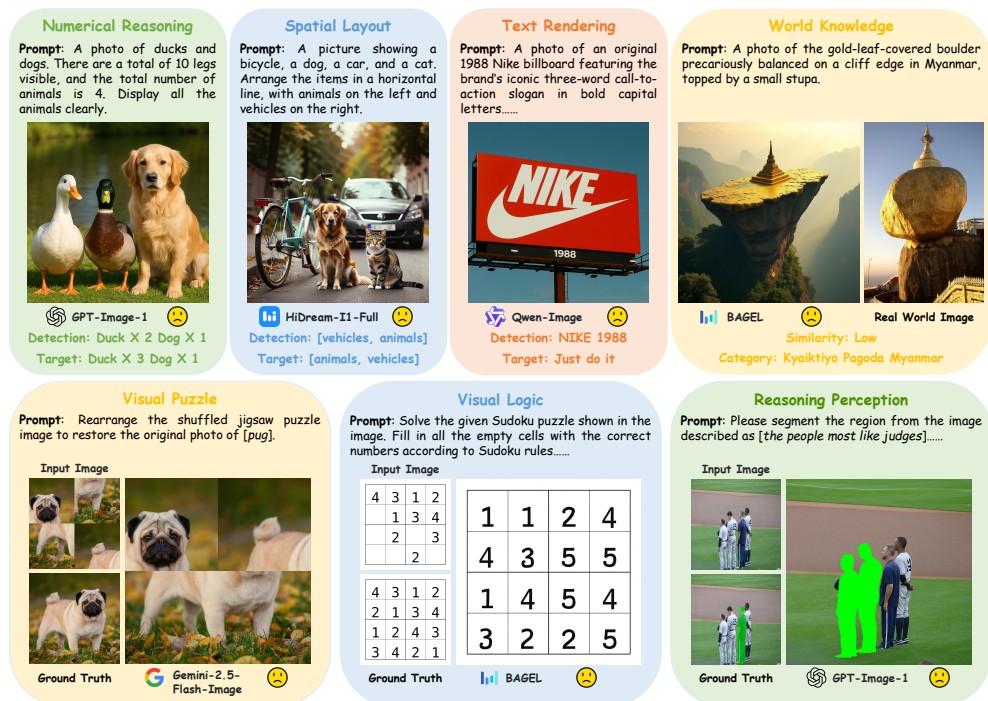

Figure 2: Examples of leading models on the GIR-Bench. Designed complex and various tasks pose challenges to current models.

carefully crafted prompts spanning three dimensions: numerical reasoning, spatial layout, and text rendering. 3) **GIR-Bench-Edit**: We further evaluate whether models can perform global planning and reasoning-driven local modifications (*c.f.*, Figure 1(c)). We construct 370 cases spanning visual puzzles, visual logic, and reasoning perception. Each case consists of an input image and its associated ground-truth image, *i.e.*, reducing evaluation bias.

In terms of evaluation, unlike previous works that heavily rely on the MLLM-as-a-Judge paradigm, we design a series of task-specific evaluation pipelines tailored for each task. Our results demonstrate that these specialized evaluations not only provide fine-grained and interpretable assessments but also effectively mitigate the inherent biases of large multimodal models. Within GIR-Bench, we systematically evaluate 21 state-of-the-art models. Our challenging benchmark reveals the limitations of leading models in performing generation tasks that require complex visual reasoning, as shown in Figure 2. Massive results show that integrating understanding with generation enables models to perform more complex visual tasks. However, for unified models, a significant gap remains between understanding and generation. Thus, effective integration of the two is crucial for unlocking the potential of unified multimodal models. In summary, our contributions are threefold:

- We propose **GIR-Bench**, a comprehensive reasoning-centric benchmark that evaluates unified multimodal models across three perspectives.
- We design a diverse suite of tasks with task-specific evaluation pipelines that provide fine-grained and interpretable metrics, moving beyond the limitations of the MLLM-as-a-Judge paradigm.
- Through extensive experiments, we reveal both the performance gap between unified and generation-only models and the internal gap between understanding and generation.

## 2 GIR-BENCH

In this section, we introduce the main components of GIR-Bench. Section 2.1 details the data sources, dataset construction, prompt suites and evaluation pipeline of GIR-Bench-UGC. In Section 2.2, we present the task dimensions and evaluation suite of GIR-Bench-T2I. Section 2.3 elaborates on the evaluation dimensions, image sources and evaluation metrics of GIR-Bench-Edit. Finally, the experiments and the insights derived from GIR-Bench will be discussed in Section 3.

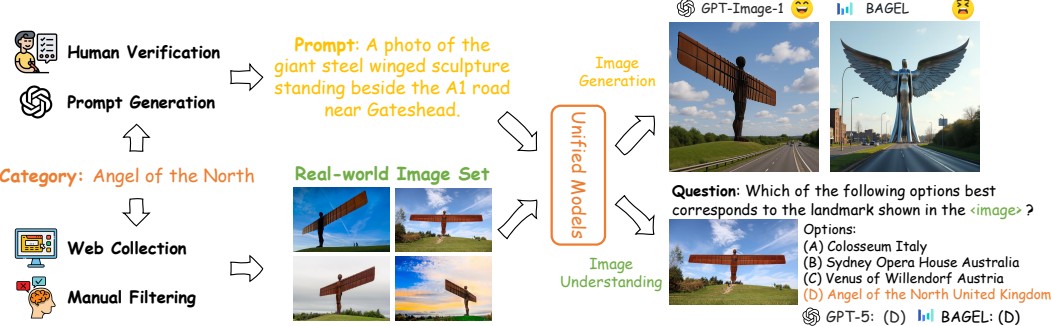

Figure 3: Illustration of GIR-Bench-UGC. For each real-world entity, an implicit prompt drives text-to-image generation, while the corresponding real image is used for image understanding evaluation.

## 2.1 GIR-BENCH-UGC

We collected 300 real-world entities from the Internet and open-source datasets, covering domains such as zoology, botany, and geography. Although previous works have explored the dimension of world knowledge, they typically remain limited to textual interpretation or text–image alignment without a deeper investigation. To evaluate reasoning ability, we utilize GPT-4o to generate implicit descriptions of each entity category, incorporating representative features such as visual appearance, historical context, and distinctive attributes. These prompts are then manually verified and refined to ensure that each uniquely corresponds to the real entity. Meanwhile, we curate a set of high-quality images to serve as reference exemplars for each entity. With entity categories, prompts, and reference image sets in place, we construct paired evaluation sets for both image understanding and image generation as shown in Figure 3. We compute the average DINOv3 (Siméoni et al., 2025) feature similarity between the generated image and the reference image sets as the evaluation metric. For image understanding, reference images are used to formulate visual question-answering cases. This design enables us to systematically investigate whether unified models can leverage the same underlying knowledge and reasoning for both understanding and generating the same entity.

## 2.2 GIR-BENCH-T2I

GIR-Bench-T2I consists of three text-to-image generation tasks that require deep reasoning. We design 300 carefully crafted prompts and construct evaluation pipelines by object detection and text recognition models. Although such problems are trivial for multimodal models, existing image generation models often fail to reason correctly and generate the expected images.

Our design of GIR-Bench is guided by three concrete principles intended to keep the benchmark objective, interpretable, and reproducible:

- **Objectivity over Subjectivity:** While many advanced reasoning scenarios (*e.g.*, causal reasoning or open-ended commonsense) typically rely on the "MLLM-as-a-Judge" paradigm, this approach often introduces bias. We prioritized tasks with deterministic solutions (*e.g.*, the unique solution in Sudoku, the original image in Jigsaw Puzzles, or exact answers in arithmetic). This ensures that our evaluation results are reproducible and indisputable.
- **Availability of Ground Truth:** We exclusively selected tasks where ground truth can be programmatically generated or strictly verified. For instance, `Spatial Layout` is verified via bounding box coordinates, and `Text Rendering` is checked via OCR. This quantifiable nature is a prerequisite for building a rigorous benchmark.
- **Focus on Implicit Reasoning & Planning:** We explicitly excluded tasks solvable by simple "keyword-to-image" mappings. The selected tasks (*e.g.*, `Visual Puzzle` and `Numerical Reasoning`) compel the model to perform implicit reasoning or global planning before generating visual content that satisfies the constraints. This is key to measuring whether a model truly comprehends the logical constraints within a prompt.

**Numerical Reasoning.** We design prompts that explicitly state mathematical constraints and record the corresponding ground truth, *e.g.*, the chicken-rabbit problem, while models must reason over these constraints to infer the correct objects and their quantities. For example, given the prompt

"*A photo of ducks and dogs. There are a total of 10 legs visible, and the total number of animals is 4. Display all the animals clearly.*", the expected output is 3 ducks and 1 dog. For evaluation, we apply object detection to extract the categories and counts of objects in the generated image and compare them against the ground truth. Notably, a case is counted as correct only when all object counts extracted from the generated image match the expected numbers. This strict criterion is necessary since the quantities of objects specified in the prompts are logically interdependent. Partial correctness indicates a broken reasoning chain, whereas full correctness ensures that the model has followed the intended reasoning process.

**Spatial Layout.** We design prompts that specify how objects should be arranged according to constraints (*e.g.,* categorical rules or ordered attributes) and record the corresponding ground truth layouts. For example, the prompt "*A picture showing a bicycle, a dog, a car, and a cat. Arrange the items in a horizontal line, with animals on the left and vehicles on the right.*" requires that the bounding boxes of animals appear to the left of those of vehicles. We evaluate results by extracting bounding boxes from generated images via object detection and verifying whether the spatial relations conform to the specified constraints.

**Text Rendering.** While existing models already demonstrate strong text rendering capabilities, their ability to reason over textual knowledge remains underexplored. To this end, we collect 60 short quotes or slogans and design implicit descriptions that correspond to them. For example, the target text "*Just do it*" corresponds to the prompt "*A photo of an original 1988 Nike billboard featuring the brand's iconic three-word call-to-action slogan in bold capital letters.*" For evaluation, we first extract text from generated images. Unlike traditional settings, our prompts are deliberately implicit, which often leads models to generate the target text along with additional irrelevant content. Common metrics (*e.g.,* word accuracy and edit distance) are not suitable, since they wrongly penalize extra words. Our goal is instead to assess whether the model successfully generates the ground truth text while allowing the presence of additional content. To this end, we propose the word-level continuous substring score as the primary evaluation metric. It is defined as $s_{\mathrm{wc}}(g,p) = \frac{|\mathcal{W}_{\mathrm{match}}(g,p)|}{|\mathcal{W}(g)|}$, where $\mathcal{W}(g)$ denotes the set of words in the ground truth $g$, and $\mathcal{W}_{\mathrm{match}}(g,p)$ counts the number of ground truth words that are fully covered by continuous character spans in the predicted text $p$.

## 2.3 GIR-BENCH-EDIT

GIR-Bench-Edit evaluates the image editing capabilities of models along three novel dimensions. Unlike previous works, each editing case in our benchmark includes both an input image and a corresponding ground truth image, thereby mitigating bias in evaluation.

**Visual Puzzle.** We filter real-world images collected in Section 2.1 to retain near-square, high-resolution photos (minimum side length $\geq 1024$ px and aspect ratio $< 1.2$), and resize each to a square. The processed images are partitioned into grids, and the tiles are randomly permuted such that at least half of the tile positions are altered. Given the shuffled image and the accompanying instruction, the model is required to reconstruct the original image, thereby evaluating its ability to integrate semantic understanding with spatial reasoning in order to restore both the global structure and the local coherence of natural images. For evaluation, we compute the Fréchet Inception Distance (FID) (Heusel et al., 2017) between generated images and ground truth. To facilitate comparison with other metrics, we further normalize FID to the range $[0, 1]$, where larger values indicate better reconstruction quality.

**Visual Logic.** We construct a high-quality dataset of Sudoku puzzles using a systematic generation pipeline. For ground truth solutions, we employ a constraint-propagation algorithm that maintains candidate sets for rows and columns, filling the grid iteratively with randomized choices while ensuring the validity of standard Sudoku constraints. For input puzzles, we adopt a deductive removal strategy, ensuring that each removed digit preserves the uniqueness of the solution. The puzzles and their corresponding solutions are then rendered into images. For evaluation, we apply text detection to extract the digits and their positions from generated images, and compute accuracy by comparing predictions against the ground truth.

**Reasoning Perception.** We select high-quality images from the LISA (Lai et al., 2024) dataset and filter those with small aspect ratios and resize them to a square. Editing instructions are constructed from their implicit textual descriptions. The task requires models to edit the target regions into

Table 1: Results of various multimodal understanding and generation models on GIR-Bench-UGC.

| Type | Model | Image Generation | | | | Image Understanding | | | |
|------|-------|---------|--------|-----------|---------|---------|--------|-----------|---------|
| | | Zoology | Botany | Geography | Overall | Zoology | Botany | Geography | Overall |
| Und | Qwen2.5VL-7B | - | - | - | - | 0.943 | 0.990 | **1.000** | 0.978 |
| | Qwen2.5-VL-32B | - | - | - | - | 0.951 | 0.990 | **1.000** | 0.976 |
| | GPT-5 | - | - | - | - | 0.983 | **1.000** | **1.000** | 0.994 |
| | Gemini-2.5-Flash | - | - | - | - | **0.991** | **1.000** | **1.000** | **0.997** |
| Gen | SD-3.5-Large | 0.263 | 0.163 | 0.437 | 0.288 | - | - | - | - |
| | HiDream-I1-Full | 0.298 | 0.218 | 0.617 | 0.378 | - | - | - | - |
| | FLUX.1-schnell | 0.239 | 0.197 | 0.440 | 0.292 | - | - | - | - |
| Unified | Show-o2-7B | 0.200 | 0.128 | 0.265 | 0.198 | 0.894 | 0.910 | **1.000** | 0.935 |
| | Janus-Pro-7b | 0.201 | 0.111 | 0.321 | 0.211 | 0.813 | 0.810 | **1.000** | 0.874 |
| | BLIP3o-NEXT-SFT-3B | 0.260 | 0.169 | 0.359 | 0.263 | 0.951 | 0.970 | **1.000** | 0.974 |
| | Ovis-U1-3B | 0.225 | 0.130 | 0.377 | 0.244 | 0.878 | 0.850 | **1.000** | 0.909 |
| | OmniGen2 | 0.214 | 0.218 | 0.451 | 0.294 | 0.932 | 0.940 | 0.984 | 0.952 |
| | UniPic2-Metaquery-9B | 0.269 | 0.195 | 0.440 | 0.301 | - | - | - | - |
| | UniWorld-V1 | 0.236 | 0.220 | 0.451 | 0.302 | - | - | - | - |
| | BAGEL-7B | 0.242 | 0.200 | 0.445 | 0.295 | 0.911 | 0.900 | **1.000** | 0.937 |
| | BAGEL-7B w/ CoT | 0.256 | 0.243 | 0.525 | 0.341 | 0.935 | 0.970 | **1.000** | 0.968 |
| | Qwen-Image | 0.293 | 0.319 | 0.677 | 0.429 | - | - | - | - |
| | Gemini-2.5-Flash-Image | 0.449 | 0.559 | 0.772 | 0.593 | - | - | - | - |
| | GPT-Image-1 | **0.568** | **0.700** | **0.800** | **0.689** | - | - | - | - |

green while keeping the background unchanged. Conceptually, this corresponds to segmentation, but since directly predicting binary masks is out of scope for editing models, we use this proxy formulation. Specifically, we instruct the model to segment the target region and render it in solid, fully opaque green while leaving the rest of the image unchanged. For evaluation, we extract the edited regions from the model outputs and convert them into binary masks. The procedure combines color-threshold detection centered on the target green, enhanced, with channel-ratio based seed detection obtained by comparing the input and generated images. The resulting binary mask is then compared with the ground-truth mask using Intersection-over-Union (IoU) as the evaluation metric.

# 3 EXPERIMENTAL RESULTS AND INSIGHTS

## 3.1 EXPERIMENT SETTINGS

**Evaluated Models.** We evaluate 21 representative models: **Multimodal understanding models**: Qwen2.5-VL-7B (Bai et al., 2025), Qwen2.5-VL-32B (Bai et al., 2025), GPT-5 (OpenAI, 2025a), Gemini-2.5-Flash (Comanici et al., 2025); **Image generation models**: SD-3.5-Large (Stability AI, 2024), HiDream-I1-Full (Cai et al., 2025), FLUX.1-schnell (Labs, 2024); **Image editing models**: FLUX.1-Kontext (Batifol et al., 2025), ICEdit (Zhang et al., 2025), Step1X-Edit (Liu et al., 2025); **Unified multimodal models**: Uniworld (Lin et al., 2025), UniPic2 (Wei et al., 2025), BAGEL (Deng et al., 2025), Ovis-U1 (Wang et al., 2025a), OmniGen2 (Wu et al., 2025b), Show-o2 (Xie et al., 2025b), Janus-Pro (Chen et al., 2025b), BLIP3o (Chen et al., 2025a), Qwen-Image (Wu et al., 2025a), GPT-Image-1 (OpenAI, 2025b), Gemini-2.5-Flash-Image (DeepMind, 2025).

**Implementation Details.** For object detection, we employ the grounding capability of InternVL3.5-38B (Wang et al., 2025b) to detect object categories and bounding boxes from generated images. For text detection, we use PPOCR v5 (Cui et al., 2025) to detect and recognize textual content, retaining only segments with confidence scores greater than 0.5.

## 3.2 EVALUATION ON REAL-WORLD UNDERSTANDING AND GENERATION

Table 1 reports results on GIR-Bench-UGC, which jointly evaluates real-world entity understanding and generation. The left panel shows understanding accuracy, while the right panel presents generation performance under implicit prompts. Overall, unified models outperform generation-only systems on reasoning-centric generation tasks, indicating that joint training across understanding and generation yields tangible benefits. GPT-Image-1 achieves the best performance and significantly outperforms other models, providing a strong upper bound. Unexpectedly, open-source uni-

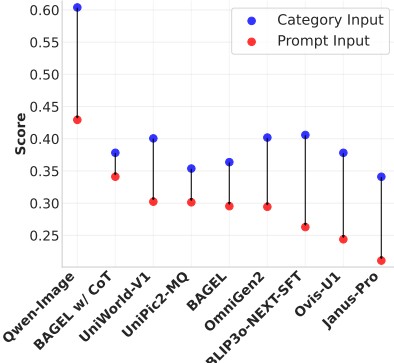

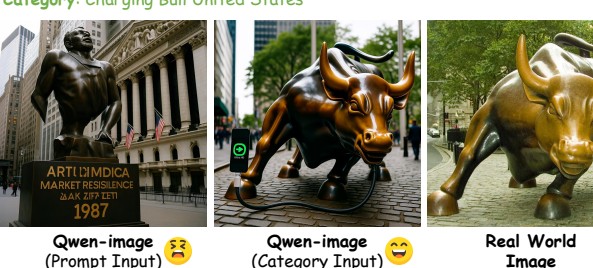

Figure 4: Performance decline from category inputs to implicit prompts.

Figure 5: Qualitative cases in GIR-Bench-UGC, showing both direct category inputs and implicit prompt inputs.

fied models do not show a clear advantage in generation compared to strong generation-only models. To further probe this phenomenon, we conducted two complementary analyses:

**Knowledge and Reasoning Capacity.** We further probed whether generation failures stem from missing world knowledge or from difficulty in reasoning. To this end, we evaluated image understanding and reported the results in Table 1. Unified models exhibit consistently strong understanding: BLIP3o attains the highest overall score of 0.974, while the lowest, Janus-Pro, still achieves 0.874. Such uniformly high scores suggest that entity recognition and basic reasoning are not the main bottlenecks. Moreover, with respect to implementation, it is worth noting that UniWorld and UniPic2 do not release official image understanding scripts. Both are developed upon a frozen Qwen2.5-VL-7B, and thus their understanding capability can be reasonably approximated by the performance of Qwen2.5-VL-7B.

**Bridging Reasoning and Generation.** To disentangle the effect of reasoning, we compare two types of inputs: (i) a *category input*, which directly specifies the target entity using the template "*a photo of* {category}", and (ii) the implicit *prompt input*, which requires reasoning to infer the entity. Figure 4 presents the per-model performance under both conditions. Across all models, scores with prompt input exhibit a marked reduction compared to category input, often by a significant margin. This pattern indicates that while models are capable of rendering entities when explicitly named, their performance deteriorates when the entity must be reasoned. The limitation therefore lies not in the ability to render the object itself, but in transferring reasoned constraints into the generative process. We further illustrate this gap through qualitative case studies in Section 4.

## 3.3 EVALUATION ON REASONING-CENTRIC TEXT-TO-IMAGE GENERATION

We report results on GIR-Bench-T2I in Table 2. Unified models outperform generation-only systems on reasoning-centric generation tasks, indicating that joint training across understanding and generation yields tangible benefits. Nevertheless, even the strongest proprietary models (*i.e.,* GPT-Image-1 and Gemini-2.5-Flash-Image) are far from perfect, highlighting the limitations of current unified models in visual generation with reasoning.

Specifically, for numerical reasoning, the highest score is just 0.362 with Gemini-2.5-Flash-Image, underscoring that models are largely unable to generate correct quantities when reasoning over implicit prompts. Among open-source unified models, BAGEL w/ CoT exhibits a substantial gain over its base counterpart (0.249 vs. 0.057), suggesting that explicit chain-of-thought helps transfer arithmetic constraints into the generative process. In spatial Layout, proprietary models again dominate, with Gemini-2.5-Flash-Image at 0.759. Within open-source unified models, BAGEL w/ CoT (0.448) outperforms Qwen-Image (0.368) and Ovis-U1 (0.356), indicating that explicit planning improves global arrangement beyond local object placement. For text rendering, under implicit descriptions, the performance gap between proprietary and open-source models is the most pronounced: GPT-Image-1 (0.813) and Gemini-2.5-Flash-Image (0.806) substantially outperform all others. While many models can render text reliably with explicit input text, they struggle to produce correct outputs when the target text must first be reasoned and then generated. Notably, BAGEL w/CoT does

Table 2: Results of various multimodal models on GIR-Bench-T2I and GIR-Bench-Edit.

| Type | Model | GIR-Bench-T2I | | | | GIR-Bench-Edit | | | |
| --- | --- | --- | --- | --- | --- | --- | --- | --- | --- |
| | | Numerical Reasoning | Spatial Layout | Text Rendering | Overall | Visual Puzzle | Visual Logic | Reasoning Perception | Overall |
| *Gen* | SD-3.5-Large | 0.107 | 0.069 | 0.227 | 0.134 | - | - | - | - |
| | HiDream-I1-Full | 0.062 | 0.230 | 0.180 | 0.157 | - | - | - | - |
| | FLUX.1-schnell | 0.045 | 0.195 | 0.238 | 0.159 | - | - | - | - |
| *Edit* | FLUX.1-Kontext-dev | - | - | - | - | 0.045 | 0.000 | 0.271 | 0.105 |
| | ICEdit | - | - | - | - | 0.023 | 0.030 | 0.233 | 0.095 |
| | Step1X-Edit | - | - | - | - | 0.037 | 0.041 | 0.136 | 0.071 |
| *Unified* | Show-o2-7B | 0.023 | 0.035 | 0.010 | 0.023 | - | - | - | - |
| | Janus-Pro-7b | 0.062 | 0.046 | 0.006 | 0.038 | - | - | - | - |
| | BLIP3o-NEXT-SFT-3B | 0.040 | 0.322 | 0.116 | 0.159 | - | - | - | - |
| | UniWorld-V1 | 0.062 | 0.207 | 0.146 | 0.138 | 0.038 | 0.013 | 0.111 | 0.054 |
| | UniPic2-Metaquery-9B | 0.107 | 0.184 | 0.126 | 0.139 | 0.107 | 0.029 | 0.261 | 0.132 |
| | Ovis-U1-3B | 0.051 | 0.379 | 0.082 | 0.171 | 0.197 | 0.033 | 0.049 | 0.093 |
| | OmniGen2 | 0.056 | 0.207 | 0.167 | 0.143 | 0.030 | 0.051 | 0.137 | 0.073 |
| | Qwen-Image | 0.153 | 0.368 | 0.150 | 0.224 | - | - | - | - |
| | Qwen-Image-Edit | - | - | - | - | 0.229 | 0.012 | 0.234 | 0.158 |
| | BAGEL-7B | 0.056 | 0.287 | 0.163 | 0.169 | 0.131 | 0.058 | 0.104 | 0.098 |
| | BAGEL-7B w/ CoT | 0.249 | 0.460 | 0.120 | 0.276 | 0.221 | 0.050 | 0.149 | 0.140 |
| | Gemini-2.5-Flash-Image | **0.362** | **0.782** | 0.806 | **0.650** | 0.382 | **0.249** | 0.399 | 0.343 |
| | GPT-Image-1 | 0.328 | 0.724 | **0.813** | 0.622 | **0.406** | 0.201 | **0.445** | **0.351** |

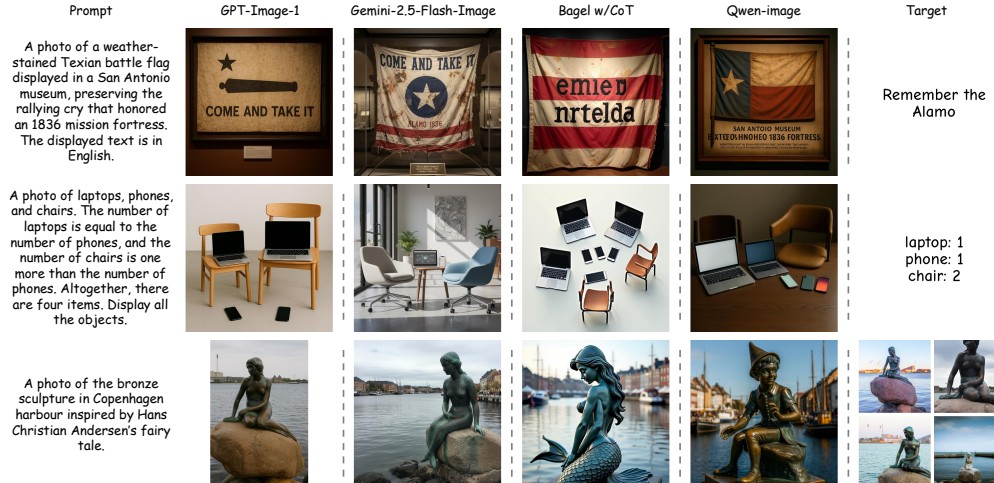

Figure 6: Illustrative examples from GIR-Bench-UGC and GIR-Bench-T2I.

not improve over BAGEL (0.120 vs. 0.163), suggesting that current reasoning traces are not yet effectively grounded in the generation process, thereby reflecting the broader gap between reasoning and generation. We further provide qualitative analysis in Section 4.

## 3.4 EVALUATION ON REASONING-CENTRIC IMAGE EDITING

Table 2 (right) presents results on GIR-Bench-Edit, which covers three editing tasks: visual puzzle, visual logic, and reasoning perception. In contrast to the trends observed in Section 3.2 and Section 3.3, the performance gaps across model categories are noticeably narrower, with all models demonstrating constrained abilities. Even the strongest proprietary models (*i.e.*, Gemini-2.5-Flash-Image and GPT-Image-1) often fail, highlighting limitations in fine-grained local control and pixel-level information maintenance for image reconstruction and region editing. While explicit reasoning mechanisms occasionally improve results, the overall effectiveness remains limited.

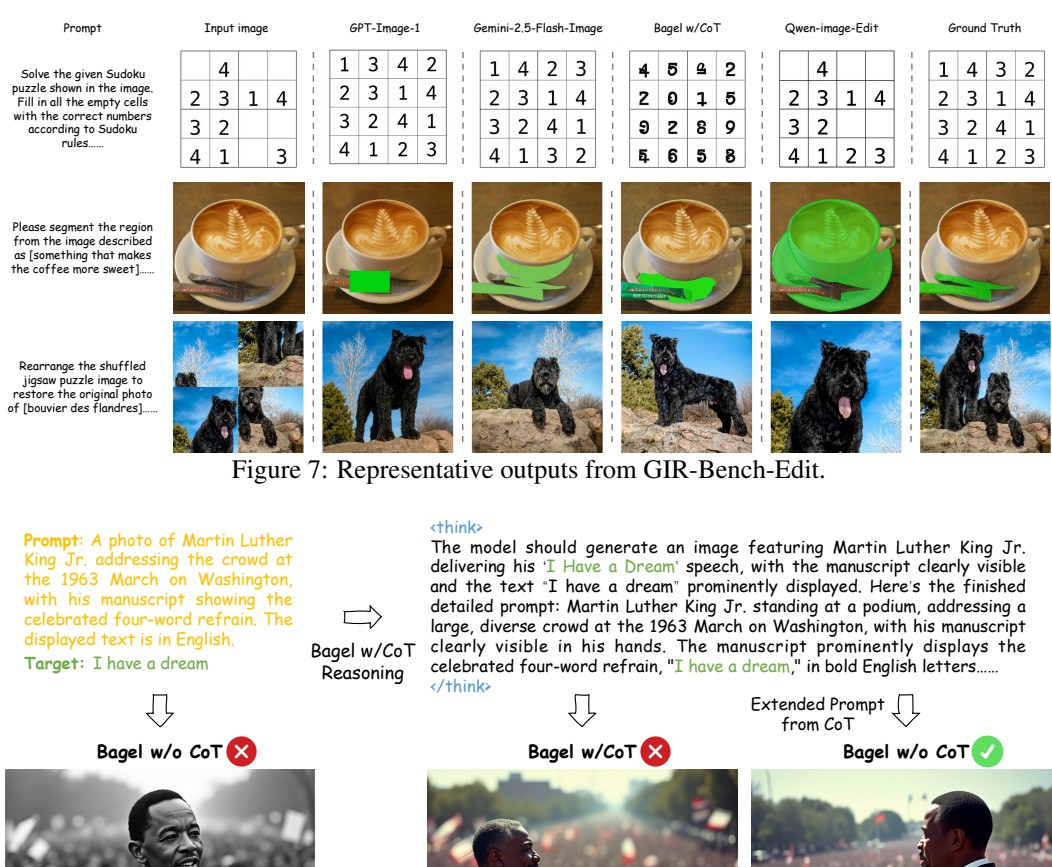

Figure 7: Representative outputs from GIR-Bench-Edit.

**Prompt:** A photo of Martin Luther King Jr. addressing the crowd at the 1963 March on Washington, with his manuscript showing the celebrated four-word refrain. The displayed text is in English.

**Target:** I have a dream

Bagel w/CoT Reasoning

<think>
The model should generate an image featuring Martin Luther King Jr. delivering his 'I Have a Dream' speech, with the manuscript clearly visible and the text "I have a dream" prominently displayed. Here's the finished detailed prompt: Martin Luther King Jr. standing at a podium, addressing a large, diverse crowd at the 1963 March on Washington, with his manuscript clearly visible in his hands. The manuscript prominently displays the celebrated four-word refrain, "I have a dream," in bold English letters......
</think>

Bagel w/o CoT ❌

Bagel w/CoT ❌

Extended Prompt from CoT

Bagel w/o CoT ✅

Figure 8: Misalignment between reasoning and generation.

## 4 QUALITATIVE ANALYSIS

**Qualitative Results.** Figure 6 and Figure 7 illustrate representative outputs across image generation and editing. For image generation, current models satisfy only part of the coupled constraints and struggle to faithfully generate correct visual content through reasoning. For image editing, visual puzzles yield globally coherent yet unfaithful reconstructions, visual logic suffers from digit errors despite clean grids, and reasoning perception reveals limitations in region preservation and editing.

**Misalignment between Reasoning and Generation.** Figure 5 presents a case study with Qwen-Image, comparing outputs under *category input* and *prompt input*. While the model produces faithful generations when the category is explicitly provided, it fails to reason correctly and generate the correct entity with the implicit prompt. To further investigate this misalignment, we analyze BAGEL w/CoT by comparing its reasoning process with the generated images. As shown in Figure 8, BAGEL correctly infers the ground-truth text (*i.e.,* "I Have a Dream") within its reasoning process and even expands it into an explicit prompt. However, when asked to generate directly from the original implicit description, it fails to render the target text. By extracting the expanded prompt from the reasoning process and re-feeding it into the model, the target text is successfully generated. This observation suggests that reasoning the target text from an implicit prompt is not the primary difficulty, and the real challenge lies in reliably transferring the inferred content into the generation.

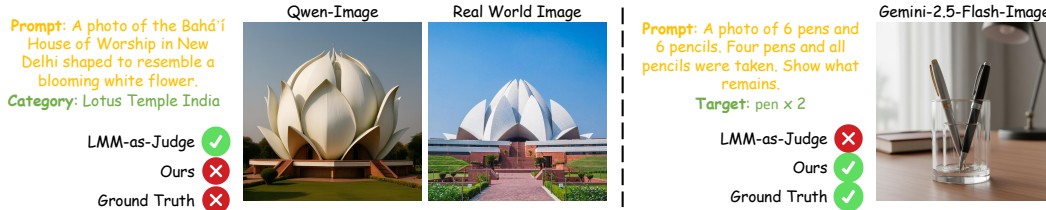

Figure 9: Comparison with MLLM-as-a-Judge.

**Qualitative Comparison with MLLM-as-a-Judge.** Figure 9 contrasts MLLM-as-a-Judge with our explicit, task-specific metrics. MLLM-as-a-Judge sometimes assigns high scores based on superficial resemblance or biased reasoning, whereas our metrics correctly penalize mismatches with the ground truth. For example, in the Lotus Temple case, judge-based scoring mislabels the Qwen-Image output as correct despite clear structural inaccuracies. In the pens-and-pencils case, it fails to reward Gemini-2.5-Flash-Image for producing the exact target. These cases highlight that our metrics yield more consistent and interpretable evaluations.

**Misalignment Understanding-Generation** Based on our quantitative ablations and qualitative case studies presented, we identify three primary factors driving the misalignment between the understanding and generation capabilities of unified models:

- **Asymmetry of Reasoning Capabilities:** Our experiments reveal a fundamental asymmetry: models often possess the knowledge but fail to "activate" it during the generation process. As illustrated in our case studies and experiments (in Figure 4 and Figure 5), the model successfully generates the correct entity when explicitly named (*e.g.*, "Charging Bull United States") but fails when the same entity must be reasoned from an implicit prompt. This suggests that the reasoning capability is localized within the LLM component. While the model correctly solves the intermediate logic, the image generator—which is optimized for visual fidelity rather than logical adherence—fails to receive or adhere to this reasoned state, leading to generation failures.
- **Information Bottleneck in Heterogeneous Architectures:** Many unified models evaluated in GIR-Bench employ heterogeneous architectures that couple a capable LLM with a separate generation head via a lightweight interface. This design creates a significant information bottleneck. While the LLM performs deep reasoning, the interface often compresses this rich semantic state into a limited number of condition tokens. Consequently, fine-grained logical constraints are often diluted or lost before reaching the pixel generation stage.
- **Lack of Interleaved Data:** Current multimodal pre-training relies heavily on static image-text pairs. These datasets map a final text description directly to an image but fail to capture the *process* of generation. Existing models lack exposure to interleaved "reasoning-trace" data (*e.g.*, `[Reasoning Step]` → `[Intermediate Visual State]` → `[Refined Image]`). Without such data, the model struggles to learn how to decompose a complex abstract instruction into a sequential plan for visual execution.

## 5 CONCLUSION

In this work, we presented GIR-Bench, a reasoning-centric benchmark that systematically evaluates unified multimodal models across understanding, generation, and editing. By grounding evaluation in explicit, task-specific metrics rather than the MLLM-as-Judge paradigm, GIR-Bench exposes fine-grained weaknesses that are otherwise hidden behind holistic scores. Extensive experiments and analyses show that while unified models consistently surpass generation-only models on reasoning-intensive tasks, they still struggle with reliably transferring reasoning into faithful visual outputs. These findings point to the importance of advancing unified models capable of seamlessly integrating reasoning and generation.

## 6 ACKNOWLEDGEMENTS

This work was supported by the Hong Kong SAR RGC General Research Fund (16219025), National Natural Science Foundation of China Young Scholar Fund Category B (62522216), National Natural Science Foundation of China Young Scholar Fund Category C (62402408), and Hong Kong SAR RGC Early Career Scheme (26208924).

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

ICLR2026

## A  ETHICS STATEMENT

This work introduces GIR-Bench, a benchmark designed to evaluate unified multimodal models on reasoning-driven image generation and editing tasks. Our study does not involve human subjects, personally identifiable information, or sensitive private data. All images used in the benchmark are either collected from publicly available open-source datasets or generated through automated pipelines, and we have carefully filtered the data to avoid inappropriate or harmful content.

We acknowledge that generative models, when misused, could produce misleading or harmful outputs. To mitigate such risks, GIR-Bench is constructed solely for research purposes, with a focus on advancing transparent and interpretable evaluation of multimodal reasoning. We explicitly avoid releasing prompts or data that could be exploited for malicious generation, and we emphasize that our benchmark should not be deployed in downstream applications without appropriate safeguards.

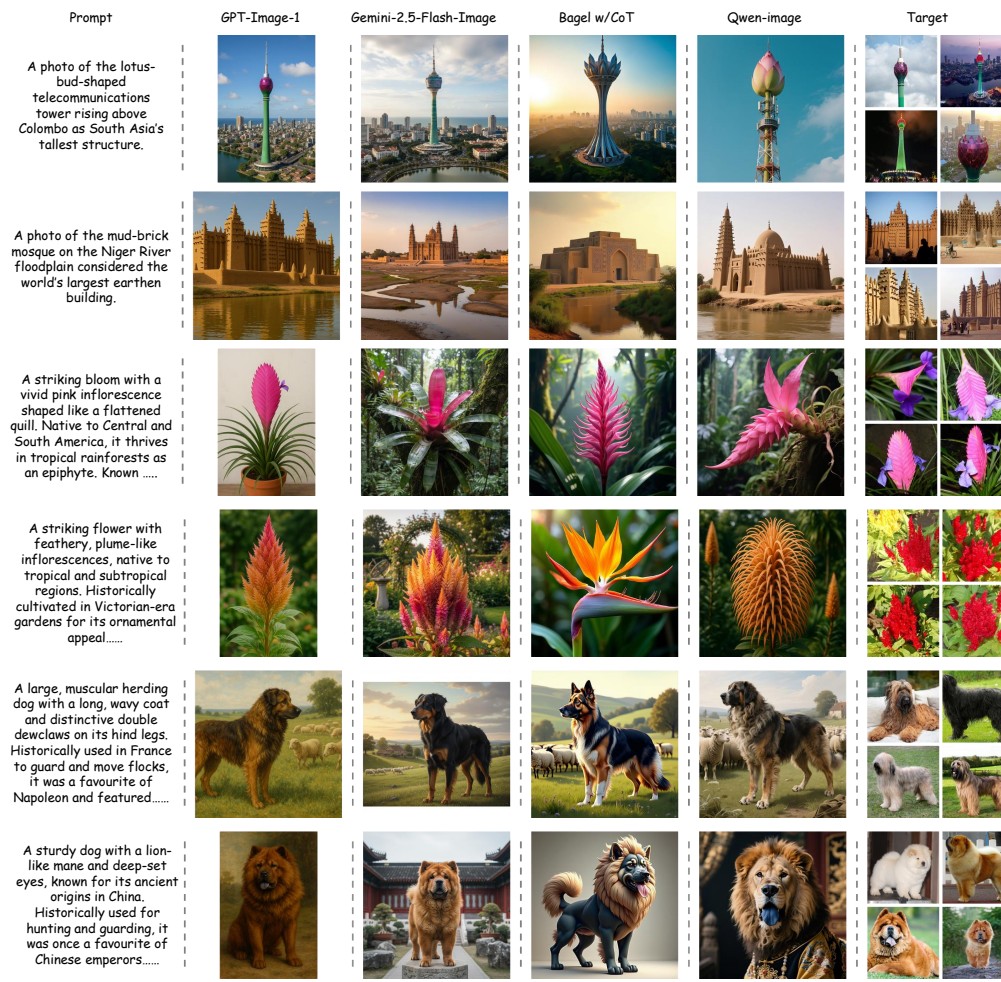

Figure 10: Examples from GIR-Bench-UGC

## B    RELATED WORK

### B.1    UNIFIED MULTIMODAL MODELS

Research in multimodal generation has shifted from modality-specific architectures to unified models that handle both understanding and generation across modalities (Lin et al., 2025; Wu et al., 2024b;a; Ma et al., 2024; Wei et al., 2025; Wu et al., 2025b; Chen et al., 2025b; Wang et al., 2025a; Zhuang et al., 2025; Xie et al., 2025a). Chameleon (Team, 2024) pioneered an early-fusion, token-based transformer that can generate and interpret text and images interchangeably, matching or surpassing proprietary systems. Emu3 (Wang et al., 2024) shows that pure autoregressive modeling can achieve general multimodal intelligence without relying on diffusion. MetaQueries (Pan et al., 2025) introduces learnable query tokens to interface a frozen multimodal LLM with a diffusion decoder, enabling knowledge-augmented image generation without degrading understanding performance. Some works attempt to integrate diffusion into unified frameworks. BLIP3-o (Chen et al., 2025a) uses a diffusion transformer to generate semantically rich CLIP image features and employs sequential pretraining (understanding followed by generation). BAGEL (Deng et al., 2025) is a decoder-only foundation model pretrained on trillions of interleaved multimodal tokens, exhibiting emergent capabilities in complex multimodal reasoning such as free-form image manipulation. Proprietary systems *e.g.,* GPT-Image-1 (OpenAI, 2025b) and Gemini 2.5 Flash Image (DeepMind, 2025), further showcase the powerful capabilities of unified multimodal models and have given rise to a wide range of real-world applications. These advancements underscore the need for comprehensive and reasoning-centric benchmarks.

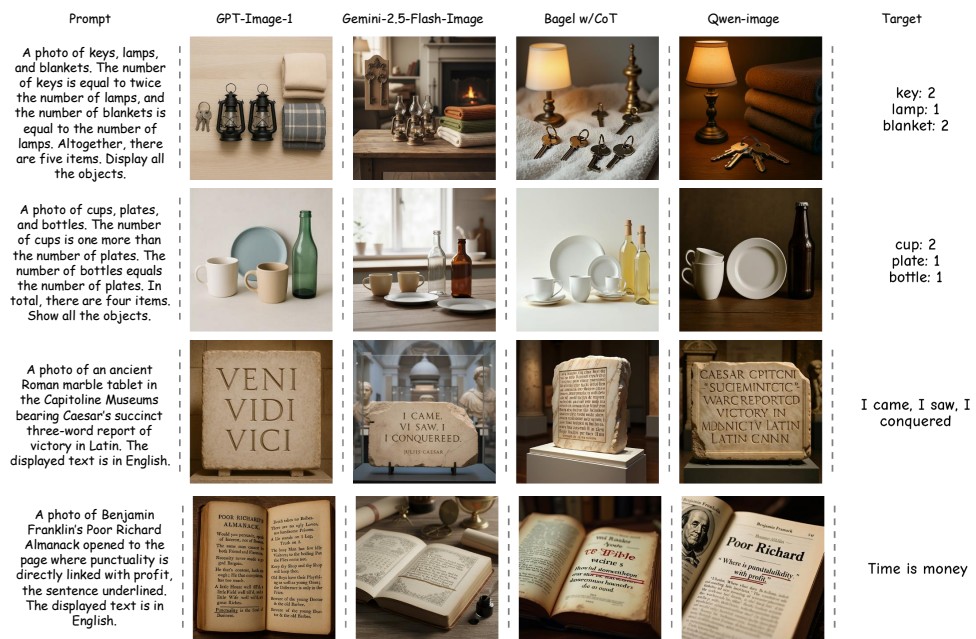

Figure 11: Examples from GIR-Bench-T2I.

## B.2 BENCHMARKS FOR UNIFIED MULTIMODAL MODELS

Existing benchmarks (Huang et al., 2023; Niu et al., 2025; Zhao et al., 2025; Li et al., 2025) for unified multimodal models cover various aspects but still fall short of probing reasoning. GenEval (Ghosh et al., 2023) uses object detection to check whether generated images match explicit constraints on object co-occurrence, position, count and colour, limiting its scope to compositional alignment. ImgEdit (Ye et al., 2025) introduces a large-scale, high-quality dataset of 1.2 million edit pairs covering both single-turn and multi-turn image editing tasks, and proposes ImgEdit-Bench, a benchmark evaluating instruction adherence, editing quality and detail preservation; however, its focus remains on surface-level editing fidelity rather than deeper reasoning. WISE (Niu et al., 2025) provides 1,000 prompts spanning cultural commonsense, spatio-temporal reasoning and natural science, and measures knowledge–image alignment with its WiScore metric, but its evaluation depends on the inherent knowledge of multimodal LLMs, making it prone to bias. T2I-ReasonBench (Sun et al., 2025) introduces 800 prompts across idiom interpretation, textual design, entity and scientific reasoning, but its two-stage evaluation (LLM-generated questions and LMM-based scoring) cannot disentangle understanding from generation. RISEBench (Zhao et al., 2025) extends evaluation to visual editing tasks involving temporal, causal, spatial and logical reasoning and scores instruction adherence, appearance consistency and visual plausibility using human and multimodal judges, yet it again depends on large models as judges rather than explicit, interpretable metrics. Current benchmarks emphasize explicit alignment and surface-level fidelity, lean heavily on multimodal models as judges, and lack a framework to decouple comprehension from generation.

## C IMAGE EDITING INSTRUCTIONS

To ensure consistency and clarity in evaluating editing capabilities, we design explicit instructions for the three tasks in GIR-Bench-Edit. Each instruction specifies the editing objective, the constraints to be preserved, and the expected output format. Placeholders such as {caption} and {category} are replaced with task-specific descriptions or categories for each case.

**Reasoning Perception**: Please segment the region from the image described as {caption}. Output the image with a solid, fully opaque green (#00FF00) segmentation mask overlay, and leave the rest of the image unchanged.

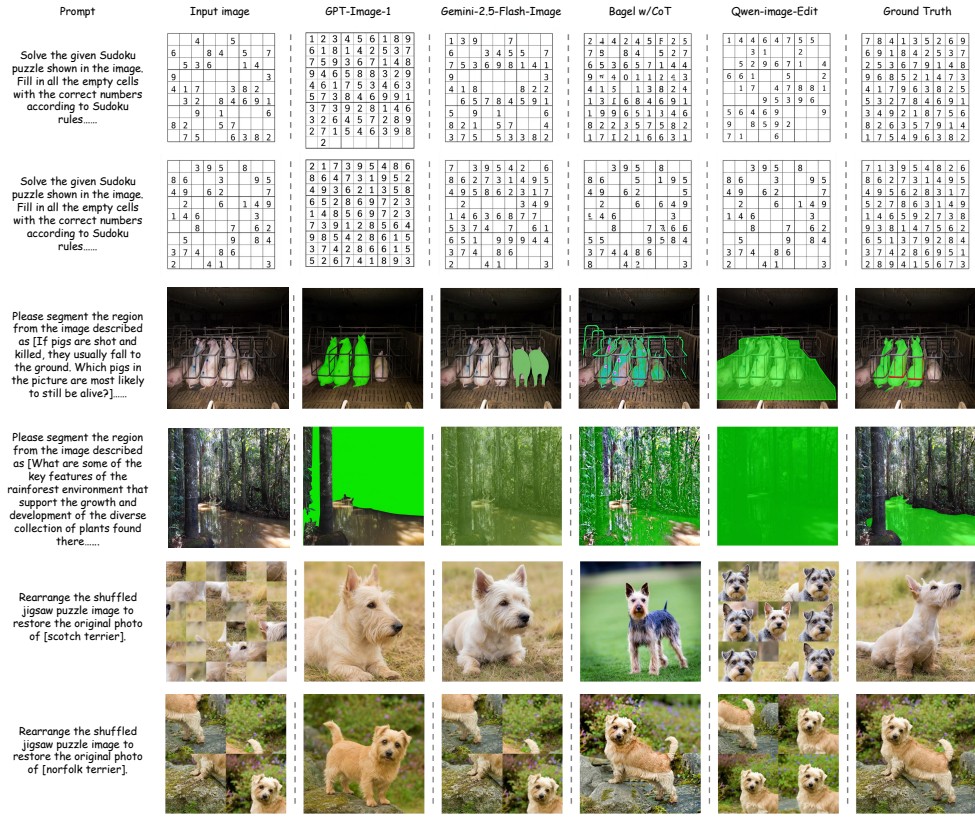

Figure 12: Examples from GIR-Bench-Edit.

**Visual Logic**: Solve the given Sudoku puzzle shown in the image. Fill in all the empty cells with the correct numbers according to Sudoku rules. Keep the existing numbers and the grid lines unchanged. Write the solution numbers in a consistent font and size, aligned within each cell, so that the final image shows the completed Sudoku solution.

**Visual Puzzle**: Rearrange the shuffled jigsaw puzzle image to restore the original photo of `{caption}`.

# D MORE QUALITATIVE RESULTS

In this section, we provide representative qualitative results from the three components of GIR-Bench. Figure 10 shows examples from GIR-Bench-UGC, Figure 11 from GIR-Bench-T2I, and Figure 12 from GIR-Bench-Edit. These cases illustrate typical challenges faced by current models and highlight the reasoning–generation misalignment observed across different tasks.

# E ROBUSTNESS AND VALIDATION OF AUTOMATED EVALUATION METRICS

To ensure the reliability of GIR-Bench, we conducted a comprehensive analysis of the robustness of our automated evaluation metrics compared to human judgment and MLLM-based judges. As illustrated in Figure 13, our automated metrics demonstrate a high Pearson correlation of $\rho \approx 0.96$ with human annotation globally. In contrast, MLLM-as-a-Judge approaches exhibit lower consistency and a notable tendency towards score inflation, particularly in lower-performance regimes (visible as points floating significantly above the diagonal in the lower-left quadrant). This quantitative evidence supports our choice of using deterministic, verifiable metrics over stochastic MLLM judgments.

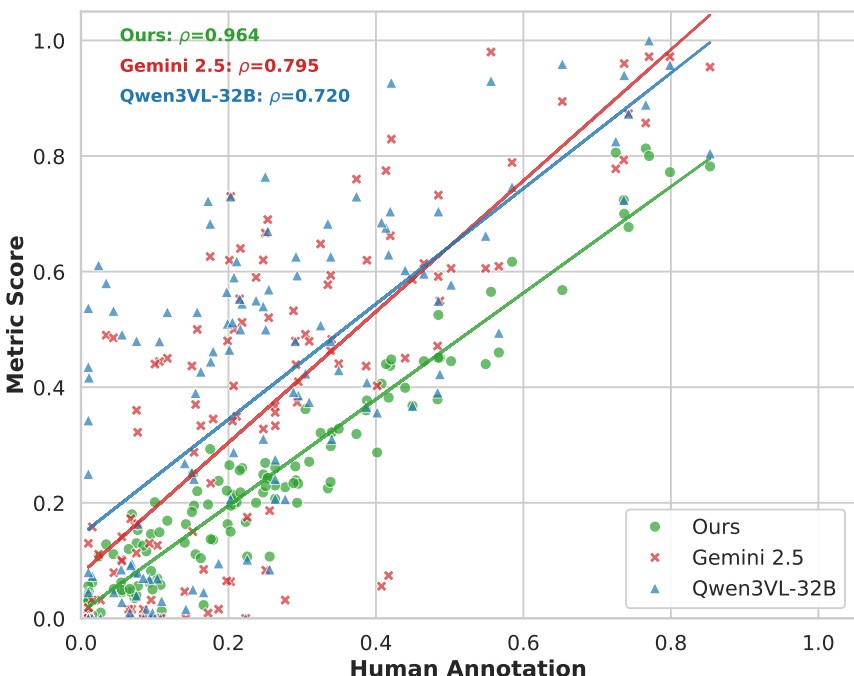

Figure 13: Correlation Analysis of Evaluation Protocols. We aggregated evaluation scores across all tasks in GIR-Bench to compare our metrics against MLLM-as-a-Judge methods. Our task-specific metrics (Green) show a strong linear correlation ($\rho \approx 0.96$) with human judgment, demonstrating high robustness. In contrast, MLLM judges (Red/Blue) exhibit higher variance and inflation bias.

We specifically investigated edge cases where automated tools might misjudge correct outputs. Our observations indicate:

- **Rare False Negatives:** Mismatches are infrequent and typically occur only when objects are rendered in highly abstract, stylized forms, are heavily occluded, or when text is severely distorted beyond standard legibility. In such cases, human annotators often also find the content ambiguous.
- **Ranking Stability:** Crucially, these edge cases are uniformly distributed and do not disproportionately affect specific models. Consequently, the relative ranking of models remains consistent. The significant performance gap is robust to these minor metric fluctuations.

## F  EXTENDED DISCUSSION: THE GAP BETWEEN UNDERSTANDING AND GENERATION

In this section, we provide an in-depth analysis of the performance gap observed between the understanding and generation capabilities of unified multimodal models. Based on our quantitative ablations in Figure 4) and qualitative case studies in Figure 5), we identify the root causes of this misalignment and propose potential research directions to bridge this gap.

### F.1  ROOT CAUSES OF THE PERFORMANCE GAP

We identify three primary factors driving the misalignment where models fail to transfer their reasoning capabilities into faithful visual generation:

- **Asymmetry of Reasoning Capabilities:** Our experiments reveal a fundamental asymmetry: models often possess the requisite knowledge but fail to "activate" it during the generation process. As illustrated in our case study on the "Charging Bull" (as shown in Figure 5), the model successfully generates the correct entity when explicitly named (*e.g.*,

"Charging Bull United States") but fails when the same entity must be reasoned from an implicit prompt. This suggests that the reasoning capability is primarily localized within the Large Language Model (LLM) component. While the model correctly solves the intermediate logic (*e.g.*, identifying the entity), the image generator—which is optimized for visual fidelity rather than logical adherence—fails to receive or adhere to this reasoned state, leading to generation failures.

- **Information Bottleneck in Heterogeneous Architectures:** Many unified models evaluated in GIR-Bench (*e.g.*, BAGEL (Deng et al., 2025), Qwen-Image (Wu et al., 2025a)) employ heterogeneous architectures that couple a highly capable autoregressive LLM with a separate generation head via a lightweight interface. This design creates a significant information bottleneck. While the LLM performs deep reasoning, the interface often compresses this rich semantic state into a limited number of condition tokens or a concise embedding. Consequently, fine-grained logical constraints (*e.g.*, exact counts in Numerical Reasoning tasks) are often diluted or lost before reaching the pixel generation stage.

- **Lack of Process-Oriented Training Data:** Current multimodal pre-training relies heavily on static image-text pairs. These datasets map a final text description directly to an image but fail to capture the *process* of generation. Existing models lack exposure to interleaved "reasoning-trace" data (*e.g.*, [Reasoning Step] → [Intermediate Visual State] → [Refined Image]). Without such data, the model struggles to learn how to decompose a complex abstract instruction into a sequential execution plan for visual generation.

## F.2 FUTURE DIRECTIONS

To narrow the gap between understanding and generation, we propose three concrete research directions supported by our empirical findings:

- **Explicit Chain-of-Thought (CoT):** Our experimental results with **BAGEL w/ CoT** (Table 2) demonstrate that externalizing the reasoning process significantly boosts performance. For instance, accuracy on the Numerical Reasoning task improves from **0.057** to **0.249**. This confirms that forcing the model to output an explicit textual plan acts as a crucial bridge, converting implicit logic into explicit constraints that the generator can better attend to and execute.

- **Native Unified Architectures:** We argue that future work should move towards natively unified paradigms where text and image tokens are treated equally within a single backbone. This would effectively eliminate the heterogeneous interface bottleneck, allowing the visual generation process to attend directly to the full, uncompressed reasoning states of the MLLM.

- **Dataset Evolution:** We advocate for the construction of reasoning-trace datasets. Future benchmarks and training sets should provide not only the final ground-truth image but also the logical execution trace. This would enable models to be supervised directly on the reasoning-to-generation mapping, rather than relying solely on end-to-end alignment.

## G CLARIFICATION FOR THE TEXT RENDERING METRIC ($s_{wc}$)

We illustrate the robustness of our proposed metric ($s_{wc}$) compared to standard accuracy ($s_{acc}$) using the Ground Truth: **"Make It Happen"**.

Table 3: Comparison of metric behaviors under different generation scenarios. Ground Truth: **"Make It Happen"**.

| Case | Generated Text (OCR Output) | $s_{wc}$ (Ours) | $s_{acc}$ |
|---|---|---|---|
| A (Perfect) | "Make It Happen" | **1.00** | 1.00 |
| B (Extra Words) | "Poster says xxxxx **Make It Happen**" | **1.00** | $\approx 0.00$ |
| C (Merged) | "**MakeItHappen**" | **1.00** | 0.86 |
| D (Missing) | "Make **It**" | **0.67** | 0.50 |

The cases in Table 3 demonstrate the specific advantages of $s_{wc}$:

- **Case B (Robustness to Context):** Unified models often generate descriptive text or artifacts (*e.g.*, "Poster says xxxxx") alongside the target. This effectively handles the "chatty" nature of these models. $s_{acc}$ fails due to the length penalty imposed by the extra characters, whereas $s_{wc}$ correctly credits the presence of the valid content.
- **Case C (Robustness to Layout):** Tight visual layouts or artistic fonts often lead to merged OCR outputs. $s_{wc}$ successfully identifies the legible words within the merged string, while $s_{acc}$ penalizes the missing whitespace.
- **Case D (Penalty for Missing):** Both metrics correctly penalize missing content. It is important to note that $s_{wc}$ acts as a strict filter: partial matches (*e.g.*, "Make It Hap") are not credited, ensuring that the metric does not reward incomplete generations.

