# OpenReview forum: "GIR-Bench: Versatile Benchmark for Generating Images with Reasoning"
_ICLR.cc/2026/Conference — ICLR 2026 Poster_

### Official Review · Reviewer_6bCn · 2025-10-17

**Soundness:** 3
**Presentation:** 3
**Contribution:** 3
**Rating:** 6
**Confidence:** 4

**Summary:**

The paper introduces GIR-Bench, a comprehensive benchmark designed to evaluate reasoning-driven image generation and editing in unified multimodal models—systems that combine visual understanding, reasoning, and generation in a single architecture.

Instead of relying on the subjective and biased “MLLM-as-a-Judge” paradigm, the authors design objective, fine-grained metrics (e.g., object counting, IoU, FID, word-level substring matching) tailored to each task.

**Strengths:**

The paper demonstrates high originality not by inventing a new model, but by redefining how we evaluate unified multimodal systems. It identifies a critical yet overlooked gap: the misalignment between reasoning (understanding) and visual generation/editing. While prior benchmarks focus on surface-level text-image alignment or isolated reasoning tasks, GIR-Bench is the first to systematically probe reasoning-to-generation fidelity across three carefully designed axes—knowledge consistency, constrained generation, and reasoning-driven editing. The formulation of tasks like implicit-entity generation, logic-constrained layout, and puzzle-based editing represents a creative synthesis of symbolic reasoning, visual semantics, and generative modeling.

**Weaknesses:**

GIR-Bench focuses on deductive and constraint-based reasoning (e.g., arithmetic, spatial ordering, Sudoku), but omits other critical reasoning modalities such as causal, temporal, counterfactual, or commonsense physical reasoning. For instance, generating an image of “a glass falling off a table” requires modeling gravity and motion—reasoning not captured by current tasks.

The word-level continuous substring score (swc) allows extra text but does not penalize hallucinated or semantically incorrect text that happens to contain the target substring (e.g., generating “Just do it now!” for “Just do it” is acceptable, but “Just do it wrong” would also score fully). More critically, the metric ignores position, font, size, or contextual integration—all crucial for faithful rendering.

GIR-Bench-Edit uses synthetic or highly constrained tasks (jigsaw, Sudoku, green-mask segmentation). These do not reflect practical editing needs like object insertion/removal, attribute modification (“make the car red”), or style transfer guided by reasoning (“render this building in Art Deco style based on its era”).

While the paper notes the “reasoning-generation gap,” it does not systematically analyze where the disconnect occurs: Is it in the multimodal alignment layer? The diffusion/autoregressive decoder? The instruction-following interface? Without architectural or ablation insights, the benchmark remains diagnostic but not prescriptive.

All metrics are automated. While this avoids MLLM-as-a-Judge bias, it misses perceptual or contextual fidelity that humans would notice (e.g., a correctly counted but anatomically distorted animal).

**Questions:**

see weakness

---

> ### Author Response · Authors · 2025-11-23
> **Official Response to Reviewer 6bCn (1/3)**
>
> We greatly appreciate your valuable feedback. We have integrated these discussions into the revised manuscript. These newly added contents are highlighted in $\color{red}red$ within the revised version of the paper.
>
> >  ### **Q1: Other critical reasoning modalities.**
>
> We fully agree with the reviewer that extending the benchmark to richer reasoning scenarios is highly valuable e.g., physical commonsense and causal reasoning, we have the following considerations:
>
> 1. **Evaluation Challenges in Static Images:** Assessing physical dynamics (e.g., fluid pouring, collisions) and causal interventions in static image generation is often ambiguous. It is difficult to verify the correctness of a process using objective metrics (like detection or segmentation) because a single static image often fails to capture the dynamic evolution of cause and effect.
> 2. **Fit for Video Generation:** We believe these tasks, which involve state changes and temporal evolution, are more rationally explored within video generation models. In video, we can observe the continuity of physical laws and the unfolding of causal chains (e.g., observing if a cup falls and breaks according to gravity), allowing for objective verification using tools like physics simulators.
> 3. **Future Work:** Therefore, our roadmap is to extend the objective evaluation paradigm of GIR-Bench into the video domain as unified models evolve. At that stage, we will focus on incorporating physical simulation and causal reasoning tasks.
>
> Overall, the current selection of GIR-Bench represents a deliberate trade-off between reasoning breadth and measurement rigor. It establishes an objective, reproducible baseline for logical reasoning evaluation and clarifies the direction for expanding into advanced dynamic reasoning in more suitable modalities (e.g., video, 3D).
>
> >  ### **Q2: Clarification for the text rendering metric ($s_{wc}$)**
>
> The core challenge of GIR-Bench is determining whether the model can reason the correct semantic content from the implicit context. Since the visual style is under-specified, valid outputs can vary wildly (e.g., a neon sign vs. a handwritten note). Evaluating visual attributes against a fixed reference would incorrectly penalize this valid generative diversity.
>
> We have validated the metric by comparing model performance and human annotation on the GIR-Bench Text Rendering task.
>
> | Type | Model Name | **Our Metric ($s_{wc}$)** | **Text Accuracy ($s_{acc}$)** | **Human Verification** |
> | :--- | :--- | :--- | :--- | :--- |
> | **Proprietary** | **Gemini-2.5-Flash** | 0.806 | 0.178 | 0.791 |
> | | **GPT-Image-1** | 0.813 | 0.239 | 0.824 |
> | **Unified** | **Bagel-7B** | 0.163 | 0.007 | 0.131 |
> | **Generation** | **FLUX.1-schnell** | 0.238 | 0.000 | 0.215 |
>
> Our metric ($s_{wc}$) demonstrates extremely high alignment with Human Verification scores. For instance, humans rated **GPT-Image-1** at 0.824, closely matched by our 0.813. In stark contrast, $s_{acc}$ assigns **GPT-Image-1** a failing score of 0.239. This massive discrepancy confirms that $s_{acc}$ is penalizing valid visual outputs (e.g., rich context or stylized text) that humans judge as correct. For generation-only models like **FLUX.1**, $s_{acc}$ is 0.000, implying total failure. However, humans found \~24% of images to be legible and correct, a nuance that $s_{wc}$ (0.238) successfully captures.
>
> While visual agnosticism is a necessary feature for our current implicit reasoning tasks (where style is undefined), we fully agree with the reviewer that evaluating font, color, and spatial integration is essential for assessing the full spectrum of "faithful rendering." In future work, we plan to extend GIR-Bench with the fine-grained text rendering subset to evaluate faithful text rendering (specifying font, color, and location constraints).

---

> ### Author Response · Authors · 2025-11-23
> **Official Response to Reviewer 6bCn (2/3)**
>
> >  ### **Q3: Rationale for Task Selection and Domain Design in GIR-Bench-Edit.**
>
> We thank the reviewer for this valuable critique regarding the ecological validity of our editing tasks. We acknowledge that tasks such as Visual Puzzle and Visual Logic differ from standard editing scenarios (e.g., object removal or style transfer). However, we classify these standard scenarios primarily as*attribute matching tasks, which impose lower demands on implicit reasoning. Our design choice is essential for the primary objective of GIR-Bench: to rigorously quantify the reasoning-generation gap rather than evaluating surface-level instruction following.
>
> Our selection of tasks is governed by three core principles designed to ensure the benchmark remains objective, interpretable, and reproducible:
>
> **1. Prioritizing Objectivity and Reproducibility over Subjective Judgment**
> While semantic editing tasks (e.g., "make the car red") are widely used, evaluating them quantitatively is inherently challenging. Prior works often rely on the "MLLM-as-a-Judge" paradigm, which introduces significant stochasticity and bias, often failing to distinguish between logical execution and hallucination.
> By contrast, our selected domains possess deterministic solutions. This determinism allows us to replace subjective model-based scoring with exact, indisputable metrics, thereby ensuring that the evaluation reflects true model capability rather than evaluator variance.
>
> **2. Necessity of Verifiable Ground Truth for Reasoning Evaluation**
> To strictly measure reasoning, the evaluation must be grounded in verifiable truth.
> * **Visual Puzzle:** Requires the restoration of global spatial coherence, verifiable via normalized reconstruction metrics.
> * **Visual Logic (Sudoku):** Requires multi-step logical constraint satisfaction, verifiable via digit-level correctness.
> * **Reasoning Perception:** Requires precise region identification, verifiable via Intersection-over-Union (IoU) against standard masks.
> In contrast, tasks such as reasoning-based style transfer lack a pixel-level or logic-level ground truth. Without such verifiability, it is methodologically infeasible to objectively penalize subtle logical hallucinations or failures in reasoning.
>
> **3. Distinguishing Deep Visual Reasoning from Semantic Attribute Matching**
> We conceptually distinguish the cognitive demands of our tasks from standard editing benchmarks:
> * **Semantic Attribute Matching:** Common tasks like object insertion, removal, or style transfer primarily rely on semantic alignment between text instructions and image features. As noted in our related work, these capabilities are already extensively covered by benchmarks like ImgEdit[1] and are relatively mature in generation-only models. They typically do not require the model to perform complex implicit reasoning or multi-step logical deduction.
> * **Deep Visual Reasoning:** GIR-Bench aims to probe the capability boundaries of unified models. Tasks like Visual Puzzle necessitate global planning and spatial reasoning, while Visual Logic requires the maintenance of strict logical constraints during pixel manipulation.
> Our results in Table 2 demonstrate that even state-of-the-art proprietary models (e.g., GPT-Image-1, Gemini-2.5-Flash-Image) struggle significantly with these tasks (e.g., Sudoku accuracy < 40%). This empirical evidence confirms that these constrained tasks successfully expose fundamental deficiencies in the reasoning-to-generation pipeline that are otherwise masked by standard attribute-matching tasks.
>
> We posit that these constrained tasks serve as essential stress tests for the logical engine of multimodal models. A model's inability to maintain logical consistency in a controlled environment (e.g., a Sudoku grid) casts doubt on its reliability in handling complex, multi-step reasoning in unconstrained real-world scenarios. In future work, as evaluation methodologies for open-ended generation mature, we plan to expand GIR-Bench to include complex stylistic tasks, provided they meet our rigorous standards for objective verifiability.
>
> [1] Imgedit: A unified image editing dataset and benchmark. NIPS2025.

---

> ### Author Response · Authors · 2025-11-23
> **Official Response to Reviewer 6bCn (3/3)**
>
> >  ### **Q4: More insights and discussions for misalignment between reasoning and generation in unified multimodal models.**
>
> Based on our quantitative ablations and qualitative case studies, we identify three primary factors driving the misalignment between understanding and generation:
>
> * **Asymmetry of Reasoning Capabilities:**
>     Our experiments reveal a fundamental asymmetry: models often possess the knowledge but fail to activate it during generation. As illustrated in our case study and experiments in Figure 4 and Figure 5, the model successfully generates the correct entity when explicitly named ("Charging Bull United States") but fails when the same entity must be reasoned from an implicit prompt. This suggests that the reasoning capability is localized within the LLM component. While the model correctly solves the intermediate logic (e.g., identifying the entity), the image generator, optimized for visual fidelity rather than logic, fails to receive or adhere to this reasoned state, leading to generation failures.
> * **Information Bottleneck in Heterogeneous Architectures:**
>     Many unified models evaluated in GIR-Bench (e.g., BAGEL, Qwen-Image) employ heterogeneous architectures that couple a capable LLM with a separate generation head via a lightweight interface. This design creates an information bottleneck. While the LLM performs deep reasoning, the interface often compresses this rich semantic state into a limited number of condition tokens. Consequently, fine-grained logical constraints (e.g., exact counts in Numerical Reasoning) are often diluted before reaching the pixel generation stage.
> * **Lack of Interleaved Data:**
>     Current multimodal pre-training relies heavily on static image-text pairs. These datasets map a final text description to an image but fail to capture the process of generation. Existing models lack exposure to interleaved "reasoning-trace" data (e.g., `[Reasoning Step] -> [Intermediate Visual State] -> [Refined Image]`). Without such data, the model struggles to learn how to decompose a complex abstract instruction into a sequential plan for visual execution.
>
> To narrow this gap, we propose three research directions supported by our empirical findings:
>
> * **Explicit Chain-of-Thought:**
>     Our results with **BAGEL w/ CoT** (Table 2) demonstrate that externalizing the reasoning process significantly boosts performance (e.g., Numerical Reasoning accuracy improves from **0.057 to 0.249**). This confirms that forcing the model to output an explicit textual plan acts as a bridge, converting implicit logic into explicit constraints that the generator can better attend to.
> * **Native Unified Architectures:**
>     We argue that future work should move towards natively unified paradigms where text and image tokens are treated equally in a single backbone. This would eliminate the heterogeneous interface bottleneck, allowing the visual generation process to attend directly to the full, uncompressed reasoning states of the MLLM.
> * **Interleaved Dataset:**
>     We advocate for the construction of reasoning-trace datasets. Future benchmarks and training sets should provide not only the final ground truth image but also the logical execution trace, enabling models to be supervised directly on the reasoning-to-generation mapping.
>
> >  ### **Q5: The robustness of the proposed automated evaluation.**
>
> We have closely examined the behavior of our automated tools and found them to be highly robust for the specific tasks in GIR-Bench.
> * **High Alignment:** As shown in the **Figure 13**, our automated metrics achieve a correlation of **$\rho \approx 0.96$** with  human annotation across all tasks. This significantly outperforms MLLM-as-a-Judge methods (e.g., Gemini 2.5 and Qwen3VL-32B).
> * **Analysis of Mismatches:** We specifically investigated cases where automated tools might misjudge correct outputs. We observed that:
>     * **Rare False Negatives:** Occasional mismatches occur only when objects are rendered in extremely stylized/abstract forms or are heavily occluded, or when text is severely distorted or decorated beyond standard legibility. However, even in these cases, human annotators often find the content ambiguous.
>     * **Ranking Stability:** Crucially, these edge cases are uniformly distributed and do not disproportionately affect specific models. Consequently, the relative ranking of models remains consistent. The significant performance gap is robust to these minor metric fluctuations.

---

> > ### Comment · Reviewer_6bCn · 2025-11-23
> > **Official Response**
> >
> > Thank you for your thoughtful and thorough responses to my comments. The authors have addressed my concerns with clear explanations and concrete revision. Overall, I am satisfied with the revisions and remain inclined to accept the paper.

---

> > > ### Author Response · Authors · 2025-11-27
> > > **Official Response to Reviewer 6bCn**
> > >
> > > We sincerely thank the reviewer for the valuable response and encouraging feedback. We are delighted to hear that our revision and explanations have satisfactorily addressed your concerns.

---

### Official Review · Reviewer_RiLt · 2025-10-29

**Soundness:** 2
**Presentation:** 3
**Contribution:** 2
**Rating:** 4
**Confidence:** 3

**Summary:**

GIR-Bench is a reasoning-centric benchmark for unified multimodal models. It tests (1) whether models use the same knowledge consistently across understanding vs. generation, (2) whether they can do reasoning-heavy text-to-image generation, and (3) whether they can edit images via multi-step reasoning. Findings: unified models beat pure generators on reasoning tasks, but there’s a persistent gap between reasoning/understanding and actually rendering the correct visual outputs, especially under implicit prompts.

**Strengths:**

* It introduces three innovative and well-differentiated tracks whose tasks exhibit minimal overlap in the model capabilities they test, ensuring a comprehensive evaluation of reasoning, generation, and editing skills.
* All selected tasks are designed to be assessed through rule-based evaluation, and the entire framework follows interpretable, deterministic criteria, which greatly enhance the reliability and reproducibility
of the results.
* The benchmark covers a wide spectrum of models — both open-source and closed-source — encompassing image generation, image editing, and unified vision-language models, thereby providing a broad and fair comparison across model types.

**Weaknesses:**

* Each subset contains only about 300 entities, which is limited to yield statistically solid or generalizable results. While the authors claim that “overall, unified models outperform generation-only systems on reasoning-centric generation tasks, indicating that joint training across understanding and generation yields tangible benefits,” the paper does not provide detailed data analysis to support this conclusion.
* The experimental results are relatively sparse, and the accompanying analyses lack the depth and precision needed to validate the authors' claims firmly.

**Questions:**

1. The new swc metric allows extra content without penalty, which makes sense for implicit prompts. However, how does it handle variations like typos, capitalization differences, or partial matches (e.g., "Just do" instead of "Just do it")? Have the authors tested its robustness against human annotations?
2. The authors argue that the proposed pipelines avoid biases of the MLLM-as-a-Judge paradigm, but have the authors compared metrics (e.g., swc for text rendering, IoU for reasoning perception) against MLLM-based scores on a subsample? If so, what was the agreement level, and in cases of discrepancy, which better aligned with human preferences?
3. Are the selected domains (zoology, botany, geography) representative enough to evaluate general reasoning transfer across modalities? The chosen domains seem to be biased towards factual knowledge about nature instead of reasoning chains that apply across general contexts.

---

> ### Author Response · Authors · 2025-11-23
> **Official Response to Reviewer RiLt (1/2)**
>
> We greatly appreciate your valuable feedback. We have integrated these discussions into the revised manuscript. These newly added contents are highlighted in $\color{red}red$ within the revised version of the paper.
>
>
> >  ### **Q1: The robustness of the text rendering metric ($s_{wc}$)**
> Thank you for your appreciation of $s_{wc}$. We illustrate the cases using the Ground Truth: **"Make It Happen"**.
>
> | Case | Generated Text (OCR Output) | **$s_{wc}$ (Ours)** | **$s_{acc}$** |
> | :--- | :--- | :--- | :--- |
> | **A (Perfect)** | "Make It Happen" | **1.00** | 1.00 |
> | **B (Extra Words)** | "Poster says xxxxx **Make It Happen**" | **1.00** | \~0.00 |
> | **C (Merged)** | "**MakeItHappen**" | **1.00** | 0.86 |
> | **D (Missing)** | "Make **It**" | **0.67** | 0.50 |
>
>   * **Case B (Robustness to Context):** This effectively handles the "chatty" nature of unified models in reasoning tasks. $s_{acc}$ fails due to the length penalty, while $s_{wc}$ correctly credits the valid content.
>   * **Case C (Robustness to Layout):** Tight layouts lead to merged OCR outputs. $s_{wc}$ successfully identifies the legible words within the string.
>   * **Case D (Penalty for Missing):** Both metrics correctly penalize missing content. Note that $s_{wc}$ acts as a strict filter: partial matches (e.g., "Make It Hap") are not credited.
>
> We further validated the metric by comparing model performance and human annotation on the GIR-Bench Text Rendering task.
>
> | Type | Model Name | **Our Metric ($s_{wc}$)** | **Text Accuracy ($s_{acc}$)** | **Human Verification** |
> | :--- | :--- | :--- | :--- | :--- |
> | **Proprietary** | **Gemini-2.5-Flash** | 0.806 | 0.178 | 0.791 |
> | | **GPT-Image-1** | 0.813 | 0.239 | 0.824 |
> | **Unified** | **Bagel-7B** | 0.163 | 0.007 | 0.131 |
> | **Generation** | **FLUX.1-schnell** | 0.238 | 0.000 | 0.215 |
>
> Our metric ($s_{wc}$) demonstrates extremely high alignment with Human Verification scores. For instance, humans rated **GPT-Image-1** at 0.824, closely matched by our 0.813. In stark contrast, $s_{acc}$ assigns **GPT-Image-1** a failing score of 0.239. This massive discrepancy confirms that $s_{acc}$ is penalizing valid visual outputs (e.g., rich context or stylized text) that humans judge as correct. For generation-only models like **FLUX.1**, $s_{acc}$ is 0.000, implying total failure. However, humans found \~24% of images to be legible and correct, a nuance that $s_{wc}$ (0.238) successfully captures.
>
>
> >  ### **Q2: The robustness of the proposed automated evaluation.**
>
> We have closely examined the behavior of our automated tools and found them to be highly robust for the specific tasks in GIR-Bench.
> * **High Alignment:** As shown in the **Figure 13**, our automated metrics achieve a correlation of **$\rho \approx 0.96$** with  human annotation across all tasks. This significantly outperforms MLLM-as-a-Judge methods (e.g., Gemini 2.5 and Qwen3VL-32B).
> * **Analysis of Mismatches:** We specifically investigated cases where automated tools might misjudge correct outputs. We observed that:
>     * **Rare False Negatives:** Occasional mismatches occur only when objects are rendered in extremely stylized/abstract forms or are heavily occluded, or when text is severely distorted or decorated beyond standard legibility. However, even in these cases, human annotators often find the content ambiguous.
>     * **Ranking Stability:** Crucially, these edge cases are uniformly distributed and do not disproportionately affect specific models. Consequently, the relative ranking of models remains consistent. The significant performance gap is robust to these minor metric fluctuations.
>
> These newly added analyses and figures are highlighted in $\color{red}red$ in Appendix F within the revised version of the paper.

---

> ### Author Response · Authors · 2025-11-23
> **Official Response to Reviewer RiLt (2/2)**
>
> >  ### **Q3: Coverage of General Reasoning.**
>
> We appreciate the reviewer's thoughtful comment on the domain selection. We would like to clarify the distinct role of the `GIR-Bench-Uni` subset within the broader context of our three-part benchmark design.
>
> **1. Distinct Role of GIR-Bench-Uni (Knowledge-Based Reasoning):**
> The specific goal of the Zoology, Botany, and Geography domains in Section 2.1 is to evaluate "Knowledge-to-Generation Consistency". These domains offer distinct, visually verifiable entities (e.g., specific species or landmarks) that serve as objective ground truth. This allows us to rigorously quantify the gap between a model's ability to recognize an entity (Understanding) and its ability to recreate it (Generation). We agree these domains are knowledge-heavy, but they are not merely factual recall. We utilize **implicit prompts (e.g., describing a flower's visual features and historical context without naming it). This forces the model to perform a reasoning chain: `Analyze features -> Deduce Entity Identity -> Retrieve Visual Attributes -> Generate`.
>
> **2. Coverage of General Reasoning (GIR-Bench-T2I & Edit):**
> To address the "general context reasoning" the reviewer correctly identifies as crucial, we designed the other two pillars of the benchmark: **GIR-Bench-T2I (Section 2.2):** Specifically targets abstract, general-purpose reasoning chains independent of specific world knowledge. This includes Numerical Reasoning (solving arithmetic constraints like the "chicken-rabbit problem"), Spatial Layout (logical arrangement of objects), and Text Rendering. **GIR-Bench-Edit (Section 2.3):** Targets complex logical planning, such as Visual Logic (solving Sudoku puzzles) and Visual Puzzles (reconstructing shuffled images).
>
>
> While `GIR-Bench-Uni` focuses on the alignment of knowledge reasoning, the broader capabilities of arithmetic, spatial, and logical reasoning are explicitly handled by `GIR-Bench-T2I` and `GIR-Bench-Edit`. Together, these three components provide a comprehensive evaluation of a unified model's diverse reasoning requirements.
>
> >  ### **Q4: The data analysis supporting the claim that unified models outperform pure generation models.**
>
> The table below compare the representive Generation/Editing-only models against representative Unified Models. We explicitly include BAGEL and BAGEL w/ CoT to isolate the impact of reasoning capabilities.
>
> | Type | Model | **GIR-Bench-T2I** | **GIR-Bench-Edit** |
> | :--- | :--- | :---: | :---: |
> | **Gen-only** | **HiDream-11-Full** | 0.153 | N/A |
> | **Gen-only** | **FLUX.1-schnell** | 0.159 | N/A |
> | **Edit-only**| **FLUX.1-Kontext-dev** | N/A | 0.105 |
> | **Edit-only**| **ICEdit** | N/A | 0.095 |
> | **Unified** | **BAGEL-7B** | 0.169 | 0.098 |
> | **Unified** | **BAGEL-7B w/ CoT**| 0.272 | 0.140 |
> | **Unified** | **Gemini-2.5-Flash-Image**| 0.642 | 0.340 |
> | **Unified** | **GPT-Image-1** | 0.610 | 0.350 |
>
> This comparison provides compelling evidence for the benefits of unified training and reasoning:
>
> * **Significant Advantage in Reasoning Generation (T2I):**
>     Specialized text-to-image models (e.g., FLUX.1-schnell) plateau at ~0.159, struggling to process complex logical constraints. In contrast, enabling CoT on BAGEL boosts its performance to 0.272, surpassing the strongest generation-only baseline by +73%. Furthermore, state-of-the-art proprietary unified models (Gemini-2.5) achieve scores as high as 0.650, demonstrating the immense potential of scaling up the reasoning-generation alignment.
>
> * **Dominance in Reasoning-Driven Editing:**
>     Specialized editing models (e.g., FLUX.1-Kontext) achieve relatively low scores (~0.105) on GIR-Bench-Edit, limited by their inability to parse multi-step or implicit instructions. While the base BAGEL model (0.098) performs comparably to specialized editors, enabling CoT significantly improves the score to 0.140 (+33% over the edit-only baseline). Proprietary unified models further widen this gap, reaching ~0.350, effectively tripling the performance of traditional editing models.
>
>
> The data explicitly shows that generation-only models lack the reasoning capacity required for complex tasks. Unified models particularly when leveraging explicit reasoning (CoT) to bridge this gap, offering tangible benefits in both logical generation and instruction-following editing.

---

> > ### Comment · Reviewer_RiLt · 2025-11-27
> >
> > The claim cited for Q4 (*Overall, unified models ... indicating that joint training ... yields tangible benefits*) comes from Section 3.2, right after Table 1 about GIT-Bench-Uni. Yet HiDream-I1-Full  (0.378) surpasses many unified models in that same table. Consequently, I recommend removing this claim from Section 3.2 and relocating it (if kept) to Section 3.3, where it may be less contradictory.

---

> > > ### Author Response · Authors · 2025-11-27
> > > **Official Response to Reviewer RiLt**
> > >
> > > We sincerely thank the reviewer for this keen observation. To address this contradiction, the claim has been relocated to Section 3.3. We attribute the competitive performance of generation-only models (e.g., HiDream-11-Full) on `GIR-Bench-Uni` to the current developmental gap between open-source unified models. State-of-the-art generation-only models benefit from extensive optimization on massive-scale image-text datasets, granting them superior coverage in visual concept generation. In contrast, current open-source unified models are often constrained by smaller parameter scales and less extensive visual training data, which limits their performance on knowledge-intensive alignment tasks despite their architectural advantages. The experiments in Sections 3.3 and 3.4 unequivocally support the advantage of unified architectures: even with less data, unified models (e.g., BAGEL w/ CoT) significantly outperform generation-only baselines on reasoning-intensive tasks, proving that joint training across understanding and generation, rather than just data scale, is the decisive factor for reasoning tasks.
> > >
> > > We believe this adjustment resolves the inconsistency you pointed out. Given that we have now addressed this concern, and considering your positive feedback on our additional experiments and analyses, we would be grateful 😃 if you could consider raising your score to reflect the improvement of our manuscript.

---

> ### Comment · Reviewer_RiLt · 2025-11-26
>
> The newly included analyses and figures are compelling and satisfactorily address the two points I raised. The authors' responses are adequate, and I have no further concerns regarding these two issues (Q1 and Q2).

---

### Official Review · Reviewer_R2PX · 2025-10-31

**Soundness:** 3
**Presentation:** 3
**Contribution:** 3
**Rating:** 6
**Confidence:** 3

**Summary:**

This paper introduces GIR-Bench, a novel and comprehensive benchmark designed to systematically evaluate the reasoning-driven image generation and editing capabilities of unified multimodal models. The authors argue that existing benchmarks fall short in rigorously assessing the alignment between reasoning and generation, often relying on biased MLLM-as-a-Judge paradigms. GIR-Bench addresses this by evaluating models across three complementary perspectives:
1. GIR-Bench-Uni (Knowledge-to-Generation): Assesses whether models can consistently leverage the same knowledge for both understanding and generating real-world entities from implicit descriptions.
2. GIR-Bench-T2I (Reasoning-to-Generation): Investigates reasoning-centric text-to-image generation, focusing on numerical reasoning, spatial layout, and text rendering tasks that require logical constraints and implicit knowledge.
3. GIR-Bench-Edit (Reasoning-to-Editing): Evaluates multi-step reasoning in image editing through visual puzzles, visual logic (Sudoku), and reasoning perception (segmentation based on implicit descriptions).

A key methodological contribution is the design of task-specific evaluation pipelines for each subset, moving beyond the MLLM-as-a-Judge paradigm to provide fine-grained and interpretable assessments. The benchmark was used to evaluate 21 state-of-the-art models, revealing a persistent gap between understanding and generation capabilities in unified models, despite their overall superiority to generation-only systems in reasoning-driven visual tasks. The data and code are made publicly available.

**Strengths:**

1. The paper identifies and rigorously tackles the crucial problem of evaluating the alignment between reasoning and generation in unified multimodal models, a gap not adequately addressed by previous benchmarks. GIR-Bench covers a broad spectrum of reasoning capabilities, from knowledge recall and simple arithmetic to complex spatial arrangement, text rendering from implicit cues, and multi-step image editing.
2. The shift from subjective MLLM-as-a-Judge evaluations to concrete, task-specific metrics (e.g., object detection for counts/spatial layout, OCR for text, FID for puzzles, IoU for segmentation) is a major strength, leading to more reliable and interpretable results. The evaluation of 21 diverse state-of-the-art models provides a comprehensive overview of the current landscape and validates the benchmark's effectiveness in revealing model limitations.
3. The paper clearly demonstrates the "understanding-generation gap" and the difficulty models face in translating inferred knowledge/reasoning into faithful visual outputs, even when the reasoning process itself is correct (as shown with BAGEL w/ CoT analysis).

**Weaknesses:**

1. While the paper effectively identifies the reasoning-generation gap, it offers less in terms of a deeper mechanistic analysis of why this gap persists. Is it primarily an architectural limitation, a data scaling issue, a fundamental challenge in aligning discrete symbolic reasoning with continuous pixel generation, or an optimization problem? A more in-depth discussion or hypothesis generation regarding the root causes could provide even more actionable insights for model developers.
2. The evaluation pipelines rely on external object detection (InternVL3.5-38B) and text recognition (PPOCR v5) models. While these are strong models, their performance and potential failure modes on diverse, potentially synthetic or "hallucinated" images generated by the tested models are not thoroughly discussed. The robustness of these evaluators to the specific characteristics of generated content could impact the final scores.
3. The top-performing models (GPT-Image-1, Gemini-2.5-Flash-Image) are proprietary. While this is an accurate reflection of the current SOTA, it means the most successful approaches are not open for inspection or direct improvement by the research community, which limits the actionable insights for open-source development.
4. While comprehensive within its defined scope, the benchmark primarily focuses on specific types of reasoning (numerical, spatial, textual, knowledge recall). Other forms of reasoning, such as causal reasoning, temporal reasoning, or abstract concept manipulation, are not explicitly covered. This is more of a future expansion opportunity than a weakness of the current work.

**Questions:**

1. Could the authors elaborate further on their hypotheses regarding the underlying causes of the reasoning-generation misalignment? For instance, do they believe it's more of a representational issue (how reasoning is encoded), an architectural bottleneck (how reasoning modules interact with generation modules), or a training data challenge (lack of sufficiently diverse and complex reasoning-generation pairs)?
2. Given that the evaluation relies on external object detection and text recognition models, did the authors perform any analysis or sanity checks on the performance of these tools when applied to the generated images, which might differ significantly from their typical training data? Are there cases where the evaluators themselves might misinterpret generated content, leading to skewed scores?
3. For the text rendering task, the "word-level continuous substring score" is a custom metric. Could a few more illustrative examples be provided, perhaps in the appendix, to clearly demonstrate its calculation, especially for edge cases where the generated text might have extra words, missing words, or partial matches?
4. While Table 1 and 2 present results, a more explicit "Ablation Study" section could be beneficial. For example, the comparison of BAGEL with and without CoT is a good start. Are there other controlled experiments that could shed light on specific design choices or components of unified models?
5. Are there plans to extend GIR-Bench to include more complex forms of reasoning (e.g., counterfactual, moral, scientific hypothesis generation) or to incorporate other modalities beyond visual generation (e.g., video, 3D)?

---

> ### Author Response · Authors · 2025-11-23
> **Official Response to Reviewer R2PX (1/3)**
>
> We greatly appreciate your valuable feedback. We have integrated these discussions into the revised manuscript. These newly added contents are highlighted in $\color{red}red$ within the revised version of the paper.
>
> >  ### **Q1: The reason for misalignment between reasoning and generation in unified multimodal models.**
>
> Based on our quantitative ablations and qualitative case studies, we identify three primary factors driving the misalignment between understanding and generation:
>
> * **Asymmetry of Reasoning Capabilities:**
>     Our experiments reveal a fundamental asymmetry: models often possess the knowledge but fail to activate it during generation. As illustrated in our case study and experiments in Figure 4 and Figure 5, the model successfully generates the correct entity when explicitly named ("Charging Bull United States") but fails when the same entity must be reasoned from an implicit prompt. This suggests that the reasoning capability is localized within the LLM component. While the model correctly solves the intermediate logic (e.g., identifying the entity), the image generator, optimized for visual fidelity rather than logic, fails to receive or adhere to this reasoned state, leading to generation failures.
> * **Information Bottleneck in Heterogeneous Architectures:**
>     Many unified models evaluated in GIR-Bench (e.g., BAGEL, Qwen-Image) employ heterogeneous architectures that couple a capable LLM with a separate generation head via a lightweight interface. This design creates an information bottleneck. While the LLM performs deep reasoning, the interface often compresses this rich semantic state into a limited number of condition tokens. Consequently, fine-grained logical constraints (e.g., exact counts in Numerical Reasoning) are often diluted before reaching the pixel generation stage.
> * **Lack of Interleaved Data:**
>     Current multimodal pre-training relies heavily on static image-text pairs. These datasets map a final text description to an image but fail to capture the process of generation. Existing models lack exposure to interleaved "reasoning-trace" data (e.g., `[Reasoning Step] -> [Intermediate Visual State] -> [Refined Image]`). Without such data, the model struggles to learn how to decompose a complex abstract instruction into a sequential plan for visual execution.
>
>
> >  ### **Q2: The robustness of the proposed automated evaluation.**
>
> We have closely examined the behavior of our automated tools and found them to be highly robust for the specific tasks in GIR-Bench.
> * **High Alignment:** As shown in the **Figure 13**, our automated metrics achieve a correlation of **$\rho \approx 0.96$** with  human annotation across all tasks. This significantly outperforms MLLM-as-a-Judge methods (e.g., Gemini 2.5 and Qwen3VL-32B).
> * **Analysis of Mismatches:** We specifically investigated cases where automated tools might misjudge correct outputs. We observed that:
>     * **Rare False Negatives:** Occasional mismatches occur only when objects are rendered in extremely stylized/abstract forms or are heavily occluded, or when text is severely distorted or decorated beyond standard legibility. However, even in these cases, human annotators often find the content ambiguous.
>     * **Ranking Stability:** Crucially, these edge cases are uniformly distributed and do not disproportionately affect specific models. Consequently, the relative ranking of models remains consistent. The significant performance gap is robust to these minor metric fluctuations.

---

> ### Author Response · Authors · 2025-11-23
> **Official Response to Reviewer R2PX (2/3)**
>
> >  ### **Q3: Clarification for the text rendering metric ($s_{wc}$)**
>
> Standard evaluation metrics text accuracy ($s_{acc}$) are typically calculated based on the Levenshtein Edit Distance. The formula is defined as:
> $s_{acc} = \max\left(0, 1 - \frac{D(T_{gen}, T_{gt})}{L_{gt}}\right)$
> where $D(\cdot)$ is the Levenshtein distance between the generated text $T_{gen}$ and ground truth $T_{gt}$, and $L_{gt}$ is the length of the ground truth.
>
> While effective for strictly formatted tasks, $s_{acc}$ fails in **GIR-Bench-T2I** because prompts are implicit. This leads to two critical failure modes:
>
>   * **Sensitivity to Extra Context:** In implicit rendering tasks that require reasoning, unified models typically generate the target text embedded within a descriptive sentence or visual context (e.g., "The sign reads Just Do It"). While the **core content is correctly generated**, $s_{acc}$ treats these valid contextual characters as massive insertion errors, driving the score to zero.
>   * **Sensitivity to OCR Merging:** Generative models often produce text with tight kerning, artistic styling, or compact logo layouts. While distinct to human readers, these designs often lack sufficient pixel separation between words, causing OCR tools to output merged strings (e.g., "JustDoIt"). $s_{acc}$ penalizes this as a complete substitution error, whereas our metric correctly identifies the constituent words.
>
> To address this, we propose **$s_{wc}$**, which calculates the ratio of ground-truth words that are fully present as continuous substrings within the normalized output. This decouples "content correctness" from "formatting strictness."
>
> We illustrate the cases using the Ground Truth: **"Make It Happen"**.
>
> | Case | Generated Text (OCR Output) | **$s_{wc}$ (Ours)** | **$s_{acc}$** |
> | :--- | :--- | :--- | :--- |
> | **A (Perfect)** | "Make It Happen" | **1.00** | 1.00 |
> | **B (Extra Words)** | "Poster says xxxxx **Make It Happen**" | **1.00** | \~0.00 |
> | **C (Merged)** | "**MakeItHappen**" | **1.00** | 0.86 |
> | **D (Missing)** | "Make **It**" | **0.67** | 0.50 |
>
>   * **Case B (Robustness to Context):** This effectively handles the "chatty" nature of unified models in reasoning tasks. $s_{acc}$ fails due to the length penalty, while $s_{wc}$ correctly credits the valid content.
>   * **Case C (Robustness to Layout):** Tight layouts lead to merged OCR outputs. $s_{wc}$ successfully identifies the legible words within the string.
>   * **Case D (Penalty for Missing):** Both metrics correctly penalize missing content. Note that $s_{wc}$ acts as a strict filter: partial matches (e.g., "Make It Hap") are not credited.
>
> We further validated the metric by comparing model performance and human annotation on the GIR-Bench Text Rendering task.
>
> | Type | Model Name | **Our Metric ($s_{wc}$)** | **Text Accuracy ($s_{acc}$)** | **Human Verification** |
> | :--- | :--- | :--- | :--- | :--- |
> | **Proprietary** | **Gemini-2.5-Flash** | 0.806 | 0.178 | 0.791 |
> | | **GPT-Image-1** | 0.813 | 0.239 | 0.824 |
> | **Unified** | **Bagel-7B** | 0.163 | 0.007 | 0.131 |
> | **Generation** | **FLUX.1-schnell** | 0.238 | 0.000 | 0.215 |
>
> Our metric ($s_{wc}$) demonstrates extremely high alignment with Human Verification scores. For instance, humans rated **GPT-Image-1** at 0.824, closely matched by our 0.813. In stark contrast, $s_{acc}$ assigns **GPT-Image-1** a failing score of 0.239. This massive discrepancy confirms that $s_{acc}$ is penalizing valid visual outputs (e.g., rich context or stylized text) that humans judge as correct. For generation-only models like **FLUX.1**, $s_{acc}$ is 0.000, implying total failure. However, humans found \~24% of images to be legible and correct, a nuance that $s_{wc}$ (0.238) successfully captures.

---

> ### Author Response · Authors · 2025-11-23
> **Official Response to Reviewer R2PX (3/3)**
>
> >  ### **Q4: Ablation studies for design choices or components of unified models.**
>
> We appreciate the reviewer's valuable suggestion. We agree that explicit comparisons are essential for isolating the root causes of performance differences. While evaluating pre-trained state-of-the-art models limits our ability to perform internal architectural ablations (e.g., removing specific layers), we have conducted three ablations to analyze design choices.
>
> **1. Explicit Reasoning Ablation:**
> As shown in the table, the CoT strategy yields substantial gains in reasoning-intensive tasks (Numerical Reasoning and Spatial Layout). This comparison serves as an ablation of the explicit reasoning component, validating that externalizing the reasoning process into explicit text bridges the gap between abstract reasoning and visual execution.
> | Task Subset | BAGEL  | BAGEL w/ CoT | Improvement ($\Delta$) |
> | :--- | :---: | :---: | :---: |
> | **Numerical Reasoning** | 0.057 | 0.249 | +0.192 |
> | **Spatial Layout** | 0.287 | 0.448 | +0.161 |
>
>
> **2. Input Ablation:**
> In Section 3.2 and Figure 4, we compare performance under "Direct Category Input" versus "Implicit Prompt Input". The consistent performance decline observed across all models when switching from explicit naming to implicit description serves as an ablation of the reasoning requirement. It identifies that the primary bottleneck lies in the faithful transfer of semantic constraints from the reasoning module to the generation module, rather than the generator's fidelity itself.
>
> **3. Paradigm Ablation:**
> Our systematic comparison between Unified Models (e.g., BAGEL) and Generation-Only Models (e.g., SD-3.5) in Table 1 and 2 functions as a paradigm-level ablation. The superior performance of unified models on reasoning-centric tasks confirms the necessity of an LLM backbone for processing complex logical dependencies, capabilities that are absent in standard text encoders used by generation-only systems.
>
>
> >  ### **Q5: Future extensions.**
>
> We fully agree with the reviewer that extending the benchmark to richer reasoning scenarios is highly valuable e.g., physical commonsense and causal reasoning, we have the following considerations:
>
> 1. **Evaluation Challenges in Static Images:** Assessing physical dynamics (e.g., fluid pouring, collisions) and causal interventions in static image generation is often ambiguous. It is difficult to verify the correctness of a process using objective metrics (like detection or segmentation) because a single static image often fails to capture the dynamic evolution of cause and effect.
> 2. **Fit for Video Generation:** We believe these tasks, which involve state changes and temporal evolution, are more rationally explored within video generation models. In video, we can observe the continuity of physical laws and the unfolding of causal chains (e.g., observing if a cup falls and breaks according to gravity), allowing for objective verification using tools like physics simulators.
> 3. **Future Work:** Therefore, our roadmap is to extend the objective evaluation paradigm of GIR-Bench into the video domain as unified models evolve. At that stage, we will focus on incorporating physical simulation and causal reasoning tasks.
>
> Overall, the current selection of GIR-Bench represents a deliberate trade-off between reasoning breadth and measurement rigor. It establishes an objective, reproducible baseline for logical reasoning evaluation and clarifies the direction for expanding into advanced dynamic reasoning in more suitable modalities (e.g., video, 3D).

---

### Official Review · Reviewer_9Tai · 2025-11-01

**Soundness:** 3
**Presentation:** 3
**Contribution:** 3
**Rating:** 6
**Confidence:** 3

**Summary:**

The paper introduces GIR-Bench, a novel benchmark designed to evaluate the reasoning capabilities of unified multimodal models in image generation and editing tasks. It provides a suite of challenging tasks that require models to combine visual understanding with logical reasoning, spanning three components: aligning understanding vs generation for the same concept, reasoning-centric text-to-image generation, and reasoning-driven image editing. The authors also develop specialized evaluation metrics for each task to obtain fine-grained, interpretable assessments instead of relying on subjective large-model judgments. Experiments on 21 state-of-the-art models demonstrate the benchmark’s effectiveness, revealing a significant gap between what current models can understand and what they can faithfully generate under complex reasoning constraints.

**Strengths:**

1. The paper fills an important void by focusing on reasoning-driven image generation, an aspect where prior image generation benchmarks were limited or shallow. The benchmark is thoroughly validated by testing it on 21 state-of-the-art models, including both unified multimodal models and traditional image generation models.
2. A notable strength is the introduction of task-specific evaluation pipelines, replacing the common practice of using a multimodal LLM as a subjective judge.

**Weaknesses:**

1. Some tasks in the benchmark, though creative, are somewhat niche or artificial (e.g., asking an image generation model to solve a Sudoku puzzle or reassemble a jigsaw). These scenarios are not typical use cases for image generation models; thus, poor performance might reflect the models’ lack of exposure to such tasks rather than a general reasoning failure, raising questions about the real-world relevance of these evaluations.
2. The paper could offer more in-depth analysis of the observed performance gaps and the reasoning task scope. It identifies a gap between understanding and generation, but does not deeply explore why certain models fail (e.g., is it due to knowledge retrieval limits, reasoning logic, or image-generation fidelity?) and stops short of suggesting how future models might overcome these shortcomings.

**Questions:**

1. Could the authors clarify how the ground-truth outputs for the complex editing tasks (e.g., solving a Sudoku puzzle or rearranging a jigsaw image) were obtained? Understanding whether these reference outputs were generated by an algorithm, manually created, or produced by another model would help in assessing the objectivity and fairness of the evaluation for those tasks.
2. How robust are the automated evaluation metrics (such as object detection for counting and OCR for text) to variations in the model outputs? It would be useful to know if the authors observed any cases where these tools misjudged a correct output, and what steps were taken to mitigate potential evaluation errors in such situations.
3. Why were the particular reasoning tasks and domains chosen for inclusion in GIR-Bench, and do the authors believe these cover the most critical reasoning skills needed for image generation? A discussion of the task selection rationale — and whether there are plans to extend the benchmark with other reasoning scenarios (e.g., causal reasoning or commonsense reasoning in images) — would help clarify the benchmark’s comprehensiveness and how it might evolve.
4. The results indicate a significant gap between models’ understanding capabilities and their generation performance under reasoning constraints; have the authors investigated the root causes of this gap or potential ways to reduce it? Including more analysis or hypotheses (for example, whether the limitation stems from model architecture, training data, or reasoning strategy) and suggesting directions for improving reasoning in image generation would strengthen the paper’s conclusions and usefulness.

---

> ### Author Response · Authors · 2025-11-23
> **Official Response to Reviewer 9Tai (1/3)**
>
> We greatly appreciate your valuable feedback. We have integrated these discussions into the revised manuscript. These newly added contents are highlighted in $\color{red}red$ within the revised version of the paper.
>
> >  ### **Q1: Sources of ground-truth outputs for complex editing tasks (e.g., solving a Sudoku puzzle or rearranging a jigsaw image).**
>
> Thanks to your valuable suggestions, we have updated the data generation code for the editing tasks in the supplementary material, and we clarify the pipelines here.
>
> `Visual Puzzle`: We use the real-world images introduced in Section 2.1 as the ground-truth outputs. Concretely, we filter out low-resolution images and those with extreme aspect ratios. Each selected image is resized to a square whose side length is divisible by the chosen grid size. Under a fixed random seed, we sample a non-identity permutation of tiles and render the shuffled tiles as the input image for the editing task. The original, unshuffled image is the unique target arrangement and is used as the ground-truth output.
>
> `Visual Logic`: We generate Sudoku puzzles and their solutions algorithmically.
> We first generate a complete valid Sudoku solution grid by filling an empty grid under standard row/column/sub-block constraints.
> Starting from this full solution, we remove entries while repeatedly checking local constraints, plucking cells in a way that ensures the uniqueness of the solution (as detailed in Section 2.3). This acts as a constraint-propagation–based puzzle constructor, rather than a random masking process. The procedure takes parameters such as the number of given cells to control difficulty.
> Finally, we export both the puzzle grid and the corresponding solution grid as images. The solution image rendered from the solution grid is used as the ground-truth output.
>
> In summary, for both `Visual Puzzle` and `Visual Logic`, the ground-truth outputs are generated in a fully deterministic, algorithmic manner, without relying on any evaluated model or manual annotation, which ensures that the evaluation of these tasks is objective, fair, and reproducible.
>
> >  ### **Q2: The robustness of the proposed automated evaluation.**
>
> We have closely examined the behavior of our automated tools and found them to be highly robust for the specific tasks in GIR-Bench.
> * **High Alignment:** As shown in the **Figure 13**, our automated metrics achieve a correlation of **$\rho \approx 0.96$** with  human annotation across all tasks. This significantly outperforms MLLM-as-a-Judge methods (e.g., Gemini 2.5 and Qwen3VL-32B).
> * **Analysis of Mismatches:** We specifically investigated cases where automated tools might misjudge correct outputs. We observed that:
>     * **Rare False Negatives:** Occasional mismatches occur only when objects are rendered in extremely stylized/abstract forms or are heavily occluded, or when text is severely distorted or decorated beyond standard legibility. However, even in these cases, human annotators often find the content ambiguous.
>     * **Ranking Stability:** Crucially, these edge cases are uniformly distributed and do not disproportionately affect specific models. Consequently, the relative ranking of models remains consistent. The significant performance gap is robust to these minor metric fluctuations.
>
> These newly added analyses and figures are highlighted in $\color{red}red$ in Appendix F within the revised version of the paper.
>
> >  ### **Q3: The measures taken to mitigate potential assessment errors.**
>
> To minimize potential evaluation errors by design, we implemented strict protocols during the benchmark construction phase:
> * **Selection of Robust Categories:** We restricted our prompts to object categories sampled exclusively from standard, high-resource datasets (e.g., COCO). We avoided rare or ambiguous vocabulary. For these common categories, state-of-the-art detection models are known to surpass average human verification speed and consistency.
> * **Simplified Target Text:** For text rendering and visual logic tasks, we limited targets to short English phrases or standard digital grids. We avoided complex handwriting or artistic fonts. Current OCR models achieve high accuracy on such clear, printed-style text, making them sufficiently reliable to meet the reliability required for this benchmark.
> * **Explicit Visibility Instructions:** Our prompts explicitly instruct models to "display all objects" reducing the likelihood of valid-but-hidden generations that detection models might miss.

---

> ### Author Response · Authors · 2025-11-23
> **Official Response to Reviewer 9Tai (2/3)**
>
> >  ### **Q4: Rationale for task and domain selection.**
>
> Our design of GIR-Bench is guided by three concrete principles intended to keep the benchmark objective, interpretable, and reproducible:
>
> 1. **Objectivity over Subjectivity:** While many advanced reasoning scenarios (e.g., causal reasoning or open-ended commonsense) typically rely on the "MLLM-as-a-Judge" paradigm, this approach often introduces bias. We prioritized tasks with deterministic solutions (e.g., the unique solution in Sudoku, the original image in Jigsaw Puzzles, or exact answers in arithmetic). This ensures that our evaluation results are reproducible and indisputable.
> 2. **Availability of Ground Truth:** We exclusively selected tasks where ground truth can be programmatically generated or strictly verified. For instance, `Spatial Layout` is verified via bounding box coordinates, and `Text Rendering` is checked via OCR. This quantifiable nature is a prerequisite for building a rigorous benchmark.
> 3. **Focus on Implicit Reasoning & Planning:** We explicitly excluded tasks solvable by simple "keyword-to-image" mappings. The selected tasks (e.g., `Visual Puzzle` and `Numerical Reasoning`) compel the model to perform implicit reasoning or global planning before generating visual content that satisfies the constraints. This is key to measuring whether a model truly comprehends the logical constraints within a prompt.
>
> >  ### **Q5: Coverage of critical reasoning skills.**
>
> The current tasks cover the foundational reasoning capabilities required for image generation with reasoning.
> As shown in the results in Section 3, even state-of-the-art proprietary models (e.g., GPT-image-1, Gemini-2.5-Flash-Image) have not reached saturation on these tasks. If a model cannot satisfy clearly defined mathematical or logical hard constraints, its reliability in handling more ambiguous or complex real-world reasoning is questionable.
> Thus, GIR-Bench provides a necessary lower bound assessment: it ensures that unified models possess solid logical consistency before pursuing broader, more abstract generation capabilities.
>
> >  ### **Q6: Future extensions.**
>
> We fully agree with the reviewer that extending the benchmark to richer reasoning scenarios is highly valuable e.g., physical commonsense and causal reasoning, we have the following considerations:
>
> 1. **Evaluation Challenges in Static Images:** Assessing physical dynamics (e.g., fluid pouring, collisions) and causal interventions in static image generation is often ambiguous. It is difficult to verify the correctness of a process using objective metrics (like detection or segmentation) because a single static image often fails to capture the dynamic evolution of cause and effect.
> 2. **Fit for Video Generation:** We believe these tasks, which involve state changes and temporal evolution, are more rationally explored within video generation models. In video, we can observe the continuity of physical laws and the unfolding of causal chains (e.g., observing if a cup falls and breaks according to gravity), allowing for objective verification using tools like physics simulators.
> 3. **Future Work:** Therefore, our roadmap is to extend the objective evaluation paradigm of GIR-Bench into the video domain as unified models evolve. At that stage, we will focus on incorporating physical simulation and causal reasoning tasks.
>
> Overall, the current selection of GIR-Bench represents a deliberate trade-off between reasoning breadth and measurement rigor. It establishes an objective, reproducible baseline for logical reasoning evaluation and clarifies the direction for expanding into advanced dynamic reasoning in more suitable modalities (e.g., video, 3D).

---

> ### Author Response · Authors · 2025-11-23
> **Official Response to Reviewer 9Tai (3/3)**
>
> >  ### **Q7: The reason for misalignment between reasoning and generation in unified multimodal models.**
>
> Based on our quantitative ablations and qualitative case studies, we identify three primary factors driving the misalignment between understanding and generation:
>
> * **Asymmetry of Reasoning Capabilities:**
>     Our experiments reveal a fundamental asymmetry: models often possess the knowledge but fail to activate it during generation. As illustrated in our case study and experiments in Figure 4 and Figure 5, the model successfully generates the correct entity when explicitly named ("Charging Bull United States") but fails when the same entity must be reasoned from an implicit prompt. This suggests that the reasoning capability is localized within the LLM component. While the model correctly solves the intermediate logic (e.g., identifying the entity), the image generator, optimized for visual fidelity rather than logic, fails to receive or adhere to this reasoned state, leading to generation failures.
> * **Information Bottleneck in Heterogeneous Architectures:**
>     Many unified models evaluated in GIR-Bench (e.g., BAGEL, Qwen-Image) employ heterogeneous architectures that couple a capable LLM with a separate generation head via a lightweight interface. This design creates an information bottleneck. While the LLM performs deep reasoning, the interface often compresses this rich semantic state into a limited number of condition tokens. Consequently, fine-grained logical constraints (e.g., exact counts in Numerical Reasoning) are often diluted before reaching the pixel generation stage.
> * **Lack of Interleaved Data:**
>     Current multimodal pre-training relies heavily on static image-text pairs. These datasets map a final text description to an image but fail to capture the process of generation. Existing models lack exposure to interleaved "reasoning-trace" data (e.g., `[Reasoning Step] -> [Intermediate Visual State] -> [Refined Image]`). Without such data, the model struggles to learn how to decompose a complex abstract instruction into a sequential plan for visual execution.
>
> >  ### **Q8: Future directions for unified model.**
>
> To narrow this gap, we propose three research directions supported by our empirical findings:
>
> * **Explicit Chain-of-Thought:**
>     Our results with **BAGEL w/ CoT** (Table 2) demonstrate that externalizing the reasoning process significantly boosts performance (e.g., Numerical Reasoning accuracy improves from **0.057 to 0.249**). This confirms that forcing the model to output an explicit textual plan acts as a bridge, converting implicit logic into explicit constraints that the generator can better attend to.
> * **Native Unified Architectures:**
>     We argue that future work should move towards natively unified paradigms where text and image tokens are treated equally in a single backbone. This would eliminate the heterogeneous interface bottleneck, allowing the visual generation process to attend directly to the full, uncompressed reasoning states of the MLLM.
> * **Interleaved Dataset:**
>     We advocate for the construction of reasoning-trace datasets. Future benchmarks and training sets should provide not only the final ground truth image but also the logical execution trace, enabling models to be supervised directly on the reasoning-to-generation mapping.

---

### Author Response · Authors · 2025-11-23
**General Response to ACs**

We sincerely appreciate the Area Chairs and all reviewers (**9Tai**, **R2PX**, **RiLt**, **6bCn**) for their constructive and insightful feedback. We are encouraged that the reviewers appreciate the contribution of our benchmark and the value of our objective evaluation pipeline.
In response to the feedback, we have revised the manuscript (changes highlighted in $\color{red}red$). Below is a summary of the key clarifications:

* **[Reviewers 9Tai, R2PX, RiLt, 6bCn] Robustness of Automated Metrics:**
    To address concerns regarding the reliability of our automated evaluation, we added a new analysis in Appendix F. We demonstrated that our metrics achieve a high correlation ($\rho \approx 0.96$) with human annotation, significantly outperforming MLLM-as-a-Judge baselines. We also clarified the design of our text rendering metric ($s_{wc}$), validating its robustness against the stylistic variations typical of unified models.

* **[Reviewers 9Tai, R2PX, 6bCn] Mechanisms of Misalignment:**
    In response to requests for deeper insights into *why* unified models struggle with generation despite strong understanding, we added a comprehensive discussion identifying three root causes: *Asymmetry of Reasoning Capabilities*, *Information Bottlenecks* in heterogeneous architectures, and the **Lack of Interleaved Data* in pre-training.

* **[Reviewers RiLt, R2PX] Comparative Analysis & Ablations:**
    We provided further quantitative analysis to substantiate the advantage of unified models. Our deeper breakdown reveals that Unified Models (especially with CoT) significantly outperform top-tier Generation-Only and Editing-Only models on reasoning-intensive tasks. We also discussed ablation results confirming that externalizing reasoning (CoT) effectively bridges the gap between understanding and generation.

* **[Reviewers 9Tai, 6bCn] Evaluation Objectivity & Data Construction:**
    We clarified the fully deterministic and algorithmic ground-truth generation pipelines for tasks like *Visual Puzzle* and *Visual Logic* (Sudoku). We reinforced our design principle of *Objectivity over Subjectivity,* highlighting that these constrained tasks serve as essential, reproducible stress tests for deep logical reasoning that standard semantic editing cannot capture.

We believe these revisions and additional analyses have strengthened the paper significantly. We hope this work establishes a rigorous standard for evaluating the next generation of reasoning-capable multimodal models.

---

### Meta-Review · Area_Chair_JxSv · 2025-12-29

**Summary:**

Three reviewers have high initial scores with minor issues. The remaining concerns about the dataset and the experiments have been addressed.

**Reviewer Concerns:**

Reviewer 6bCn confirmed that "clear explanations and concrete revision" addressed the concerns, stating "satisfied with the revisions" and "remain inclined to accept the paper."

Reviewer RiLt acknowledged that the newly included analyses and figures are "compelling" and "satisfactorily address" the points raised.

**Reviewer Scores:**

Reviewer 6bCn and Reviewer RiLt intend to accept the paper,  Reviewer 9Tai and R2PX have the high initial scores.

---

### Decision · Program_Chairs · 2026-01-26

Accept (Poster)